**Estimation of subsurface porosities and thermal conductivities of polygonal tundra by**
**coupled inversion of electrical resistivity, temperature, and moisture content data**
Elchin E. Jafarov[1], Dylan R. Harp[1], Ethan T. Coon[2], Baptiste Dafflon[3], Anh Phuong Tran[3,4],
Adam L. Atchley[1], Youzuo Lin[1], and Cathy J. Wilson[1]
1. Earth and Environmental Sciences Division, Los Alamos National Laboratory, Los Alamos,
New Mexico, USA
2. Climate Change Science Institute and Environmental Sciences, Oak Ridge National
Laboratory, Oak Ridge, Tennessee, USA
3. Climate and Ecosystem Division, Lawrence Berkeley National Laboratory, Berkeley,
California, USA
4. Department of Water Research Engineering and Technology, Water Research Institute,
Hanoi, Vietnam
**Abstract**
Studies indicate greenhouse gas emissions following permafrost thaw will amplify current rates of
atmospheric warming, a process referred to as the permafrost carbon feedback. However, large
uncertainties exist regarding the timing and magnitude of the permafrost carbon feedback, in part
due to uncertainties associated with subsurface permafrost parameterization and structure.
Development of robust parameter estimation methods for permafrost-rich soils is becoming urgent
under accelerated warming of the Arctic. Improved parameterization of the subsurface properties
in land system models would lead to improved predictions and reduction of modeling uncertainty.
In this work we set the groundwork for future parameter estimation (PE) studies by developing
and evaluating a joint PE algorithm that estimates soil porosities and thermal conductivities from
time-series of soil temperature and moisture measurements, and discrete in-time electrical
resistivity measurements. The algorithm utilizes the Model Independent Parameter Estimation and
Uncertainty Analysis toolbox and coupled hydro-thermal-geophysical modeling. We test the PE
algorithm against synthetic data, providing a proof-of-concept for the approach. We use specified
subsurface porosities and thermal conductivities and coupled models to setup a synthetic state,
perturb the parameters, then verify that our PE method is able to recover the parameters and
synthetic state. To evaluate the accuracy and robustness of the approach we perform multiple tests
for a perturbed set of initial starting parameter combinations. In addition, we varied types and
quantities of data to better understand the optimal dataset needed to improve the PE method. The
results of the PE tests suggest that using multiple types of data improve the overall robustness of
the method. Our numerical experiments indicate that special care needs to be taken during the field
experiment setup so that (1) the vertical distance between adjacent measurement sensors allows
the signal variability in space to be resolved and (2) longer time interval between resistivity
snapshots allows signal variability in time to be resolved.

## 1. Introduction

Subsurface soil property parametrization contributes to a wide uncertainty range in projected
active layer depth and in simulated permafrost distribution in the Northern Hemisphere when
predicted using Land System Models (Koven et al., 2015; Harp et al., 2016). Reduction of this
uncertainty is becoming urgent with recent accelerated thawing of permafrost (Biskaborn et al.,
2019). Warming permafrost leads to increased infrastructure maintenance costs (Hjort et al., 2018),
has a positive feedback on global climate change (McGuire et al., 2018), and increases the
probability of the potential hazards for human health (Schuster et al., 2018). Better subsurface soil
property parametrizations in Land System Models requires the development of methods that can
robustly estimate these soil properties including porosity and thermal conductivity of peat and
mineral layers.
Direct measurements of subsurface soil properties are labor intensive, destructive, and not always
feasible (Smith and Tice, 1988; Kern, 1994; Boike and Roth, 1997; Yoshikawa et al., 2004). While
soil sample analysis can provide critical information on soil properties at a fine scale, this
information is limited to sparsely sampled locations. Multiple methods used in the laboratory to
measure soil properties by using soil cores extracted from the field site are well summarized by
Nicolsky et al., (2009), but logistical and economic burden typically do not allow these
measurements to be made in the field. Inverse modeling serves as an alternative approach to
recover soil properties using a combination of indirect and direct measurements and physics-based
numerical models.
Different inverse modeling frameworks have been developed to estimate soil thermal properties
using physical-based models and time-series of ground temperature data. Some earlier studies
used heat equation models without phase change (Beck et al., 1985; Allifanov et al., 1996). More
recent works include phase change, which is an important component of the energy balance in
permafrost-affected soils (e.g. Nicolsky et al., 2007; 2009, Tran et al., 2017). Nicolsky et al.,
(2007; 2009) used an optimization based inverse method and a variational data assimilation
method to estimate soil properties. In particular, Nicolsky et al., (2007; 2009) used measured
subsurface temperatures to inversely estimate thermal conductivities, porosities, freezing point
temperatures, and unfrozen water coefficients, pointing out that sensitivity analyses (i.e.
perturbation of the parameter values) are required in order to robustly establish a set of estimated
parameters. Harp et al., (2016) used an ensemble-based method to evaluate the uncertainty of
projections of permafrost conditions in a warming climate due to uncertainty in subsurface
properties. Atchley et al., (2015) used data calibration to estimate hydrothermal properties of soils.
All these methods used ground temperatures alone to estimate soil properties and 1D soil columns
assuming a 1D soil structure.
Recently, Tran et al., (2017) used a coupled hydrological–thermal–geophysical modeling approach
to estimate soil organic content. The approach was based on coupling the 1D Community Land
Model (CLM4.5; Oleson et al., 2013) that simulates surface-subsurface water, heat and energy
exchange and the 2D Boundless Electrical Resistivity Tomography (BERT) forward model
(Rücker et al., 2006). The simulated 1D snapshots of the subsurface temperature, liquid water and
ice content from the CLM model were explicitly linked to soil electrical resistivities via
petrophysical relationships which were then used as input to BERT's forward model to calculate
apparent resistivities. Their inverse modeling framework aims to minimize the misfit between
calculated and measured data, including soil temperature, liquid water content and apparent
resistivity. Here we modify and extend this approach to 2D by using the Advanced Terrestrial
Simulator (ATS) model, which was specifically developed to study fine-scale hydrothermal
processes of permafrost-affected soils. In addition, instead of estimating organic content of the soil
as in Tran et al., (2017), we estimate porosities and thermal conductivities of peat (organic) and
mineral layers across a 2D transect within polygonal tundra.
Modeling the full, continuous 2D transect allows us to simulate lateral hydro-thermal fluxes not
possible with individual 1D columns known to be important in polygonal tundra (Abolt et al, 2018,
Liljedahl et al, 2016). At each grid cell in the transect, a physical state develops during the ATS
simulation (temperature, saturation, etc.) that is then used to calculate heterogeneous electrical
resistivities via petrophysical relations. This allows more realistic simulated apparent resistivities
that include the effects of lateral hydrothermal connectivity within the transect.
Through this approach, we develop a parameter estimation (PE) algorithm that aims to estimate
porosities and thermal conductivities in permafrost-affected soils through joint inversion of
hydrothermal and geophysical measurements, including ground temperature, saturation, and
apparent resistivity. Our main objective then is to evaluate which types and number of
measurements are necessary to constrain the inversion to yield a robust and accurate prediction of
subsurface porosities and thermal conductivities. The inverse modeling framework couples the
state-of-the-art hydrothermal permafrost simulator ATS, electrical resistivity software package
BERT and the Model Independent Parameter Estimation and Uncertainty Analysis toolbox (PEST)
software package (Doherty, 2001). We progressively test the accuracy and robustness of the
method using a series of synthetic problems by: 1) increasing the complexity of the meteorological
data used to drive the coupled thermo-hydro-geophysical model and 2) testing the inclusion of
individual and combinations of several available measurement types on the accuracy and
robustness of inversions. The results of this work can be used to better understand challenges
associated with subsurface porosity and thermal conductivity estimation. Additionally, we used
findings from this study to suggest how data should be collected to improve the accuracy of the
estimated soil properties and to optimize the total number of measurements needed to make a
robust subsurface PE.

**2. Methods**
We estimate the soil properties of porosity and soil grain thermal conductivity for peat and mineral
layers of a 2D transect within polygonal tundra. Our PE approach is summarized in Figure 1. Given
specified "true" values of these parameters, we used the ATS version 0.86 model to solve for a
transient, spatially distributed hydro-thermal state characterized by temperature and liquid and ice
saturations. ATS is a 3D-capable coupled surface and groundwater flow and heat transport model
representing the soil physics needed to capture permafrost dynamics, including flow of unfrozen
water in variably-saturated, partially-frozen, non-homogeneous soils (Painter et al., 2016). Given
this hydrothermal state, we calculate resistivity values at every grid cell via petrophysical
relationships, and run the forward modelling component of the BERT software package (Rücker
et al., 2006) to simulate resistance and related apparent resistivity values that would be measured
with ground-coupled electrodes and an ERT acquisition system.

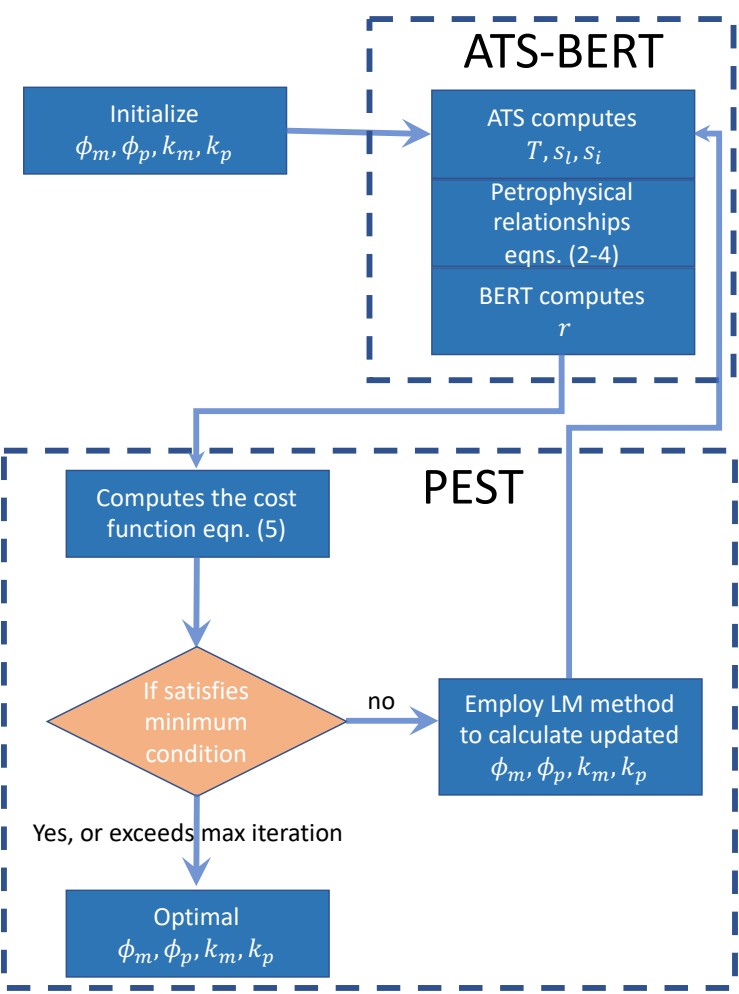


**Figure 1.** Schematics of the parameter estimation algorithm. The algorithm starts with initial guesses on
porosities and thermal conductivities $\{\phi, k\}=\{\phi_m, \phi_p, k_m, k_p\}$ for the peat and mineral layers. The coupled
ATS-BERT forward model then simulates temperature ($T$), liquid water saturation ($s_l$), ice saturation
($s_i$), and apparent resistivities ($\rho_a$), which are passed to the cost function. If the cost function is small
enough, $\{\phi, k\}$ are considered to be the estimated parameters. If not, the values of the $\{\phi, k\}$ are updated
according to the Levenberg-Marquardt (LM) minimization algorithm and passed back to the ATS-BERT
model.

**2.1 ATS-BERT Model**
To set up the synthetic model, we used digital elevation data of a transect through ice-wedge
polygonal tundra at the Barrow Environmental Observatory (BEO), at Utqiagvik, Alaska (Fig. 2).
Our study includes an 11 m section covering a single polygon with an ice-wedge on each side. In
this study we do not explicitly assign ice properties for the ice-wedges. Instead, we model bulk
porosities and effective thermal conductivities that can be associated with peat and mineral layers
of the entire transect.
In Figure 2A, we present the computational mesh representing the cross-section of the polygonal
tundra that ATS is run on. The thickness of the peat layer corresponds to observations at the site,
with a thick peat layer on the sides (troughs) and a thinner layer in the middle of the low-centered
polygon. A mineral layer was assigned below the peat layer across the transect. We initially
designated six synthetic direct temperature and soil moisture measurement locations within the
active layer area, the maximum thaw layer from the ground surface to the top of the permafrost,
similar to the sensor setup at the site (Dafflon et al., 2017). The average active layer depth is about
38cm, as it can be seen from the ground temperatures simulated for the synthetic model run with
actual meteorological data in Figure 2B. The linear white region on Fig. 2B indicates the bottom
of the active layer within the transect ($0^{\circ}$ C). Then we added four more synthetic direct
measurement locations below the active layer to evaluate the effect of their inclusion on PE
accuracy and robustness. All observation locations are represented as stars on Figure 2A
corresponding to the locations of the collected daily averaged temperature and soil moisture
timeseries. The temperature and soil moisture timeseries were recorded at depths of 5, 20, 60, and
80 cm below the surface.
The setup of the ATS model followed a standard procedure described in several studies (Atchley
et al., 2015; Painter et al., 2016; Jafarov et al., 2018). Typically, we set up the model in several
steps: 1) initialization of the water table, 2) introduction of the energy equation to establish
antecedent permafrost, and 3) spinup of the model with simplified and actual meteorological data
from the BEO station. We spun up the model until the active layer achieved cyclical equilibrium.
The overall depth of the modeling domain is 50 m. We set the bottom boundary to a constant
temperature of T=263.55K and set zero heat and zero mass flux boundary conditions on the vertical
sides. A seepage face was imposed at 4 cm below the surface on each side of the domain to allow
drainage to the trough network to prevent water from pooling at the surface, as is typical of partially
degraded polygonal ground (Liljedahl et al, 2016). We use two types of meteorological datasets
as surface boundary condition drivers for the ATS model: simplified (sinusoidal air temperature,
constant precipitation, and constant radiative forcing) and actual weather data from the BEO site.
The actual meteorological data were collected starting on January 1, 2015 and include air
temperature, rain and snow precipitation, humidity, long and shortwave radiation, and wind speed.
We created a synthetic truth by designating porosities and soil grain thermal conductivities $\{\phi, k\}$
of peat and mineral soil as parameters in the forward model. The resulting temperature ($T$), liquid
and ice-water saturations ($s_l, s_i$), and apparent resistivities ($\rho_a$) were collected as the true state.
Critical for these simulations is the calculation of the thermal conductivities of the bulk soil;
calculated in ATS using Kersten numbers to interpolate between saturated frozen, saturated
unfrozen, and fully dry states (Painter et al., 2016) where the thermal conductivities of each end-
member state is determined by the thermal conductivity of the components (soil grains, air, water,
or liquid) weighted by the relative abundance of each component in the cell (Johansen, 1977;
Peters-Lidard et al, 1998; Atchley et al., 2015). Thermal conductivities of water, ice, and air are
considered constant, leaving soil grain thermal conductivity as the remaining parameter to be
estimated. The equation to calculate saturated, frozen thermal conductivity ($\kappa_{sat,f}$) has the
following form:
$$\kappa_{sat,f} = \kappa_{sat,uf} \cdot \kappa_i^{\phi} \cdot \kappa_w^{-\phi}, \tag{1}$$

where $\kappa_{sat,uf}$, $\kappa_i$, $\kappa_w$ are thermal conductivities for saturated unfrozen, ice, and liquid water,
respectively, and $\phi$ is porosity.
The freezing characteristic curve is thermodynamically derived using a Clapeyron relation and the
unfrozen water retention curve, as described in Painter and Karra (2014) and Painter et al., (2016).
In Figure 3 we present liquid and ice saturations for one realization of the model for winter
(January) and summer (August) times of the year. The ice saturation is high below the active layer
all year long and lowest within the active layer in the summer. The peat layer holds more water;
therefore, ice concentration is higher than in the mineral layer in the winter. The liquid saturation
plot shows that by the end of the summer, the peat layer is drier than the mineral layer.
We sequentially couple the ATS and BERT numerical models via petrophysical relationships used
by Tran et al. (2017) and based on Archie (1942) and Minsley et al. (2015). In that approach, the
electrical resistivity ($\rho$) is determined as a function of soil characteristics, temperature, porosity,
liquid water saturation, fluid conductivity, and ice content:
$$\rho = 1/(\phi^d [s_l^n \sigma_w + (\phi^{-d} - 1)\sigma_s] \cdot [1 + c(T - 25)]), \tag{2}$$

where $\sigma_w$ is the fluid electrical conductivity, $\sigma_s$ is soil/sediments electrical conduction, $n$ is a
saturation index, $d$ is a cementation index, and $c$ is a temperature compensation factor accounting
for deviations from $T = 25^o C$.
The ice content is linked to water content through the liquid-water saturation and to $\sigma_w$, which is
influenced by the concentration of $Na^+$ and $Cl^-$ ions and the ice/liquid fraction. Here $\sigma_w$ has the
following form:

$$\sigma_w = \sum_{i=1}^{n_{ion}} F_c \beta_i |z_i| C_{i(S_{f_{i=0}})} S_{fw}^{-\alpha} \tag{3}$$

Where $F_c$ is Faraday's constant, $\beta_i$ and $z_i$ the ionic mobility and valence respectively, $C_i$ is the
concentration of $i^{th}$ ion, $\alpha$ is factor influencing how the liquid water salinity increases when the
fractions of liquid in ice-liquid water $S_{f_w}$ decreases. $S_{f_w}$ is defined as:

$$S_{f_w} = s_l/(s_l + s_i) \tag{4}$$

Both $s_l$ and $s_i$ are simulated by ATS. Note that $\phi$ in eq. (2) is an estimated parameter (see
Figure 1). In this study we assume that $n, d, \sigma_s, \alpha, F_c, \beta_{Na^+}, \beta_{Cl^-}, C_{Na^+}$, and $C_{Cl^-}$ parameters used
in equations (2) and (3) are known (see Tran et al., 2017) and focus on the robustness of the PE
algorithm in estimating porosity and thermal conductivity.
The 2D resistivity data inferred from ATS simulations and petrophysical relationships gets
passed to BERT which simulates resistances that are then converted to the apparent resistivities
$(\rho_a)$. The $\rho_a$ values correspond to an acquisition along an 11 m long transect using a 0.5 m
electrode spacing and a Schlumberger configuration with a total of 138 measurements (see Fig.
2B). This configuration implies that the measurements are mostly sensitive to the electrical
resistivity in the top few meters.
Since BERT and ATS operate on different unstructured meshes, we wrote a function that
interpolates the values between the two meshes. Note that the ATS mesh is 50m deep. We
calculate $\rho$ by using corresponding outputs from the ATS model and the petrophysical
relationships and then interpolated these values on a mesh defined in BERT and adapted to the
acquisition geometry. BERT's mesh consists of a finely resolved mesh (11m wide by 4.5m
deep) embedded within a coarser outer mesh that is about 120m wide and 85m deep. We link
hydrological variables with electrical resistivities in the fine mesh. The coarse mesh is used to
reduce the effect of boundaries. It extends until the change in the electrical resistivity between
two neighboring cells is negligible.

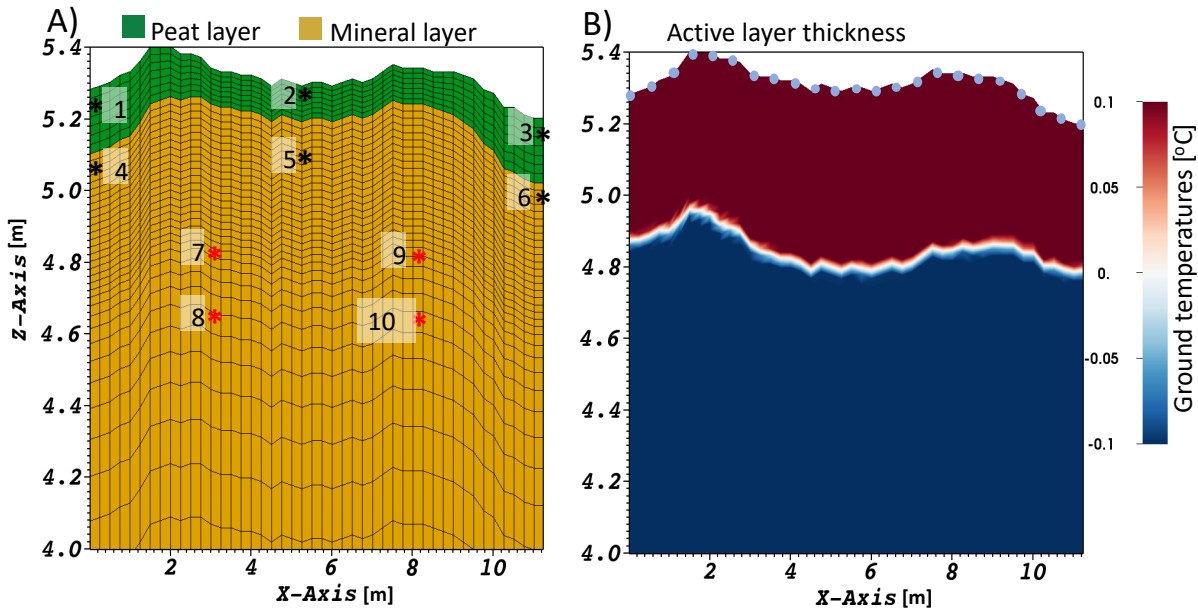


**Figure 2.** The (vertically exaggerated) 2D transect used by the ATS model. A) The unstructured mesh
where green represents the peat layer and brown represents the mineral soil layer. Black stars represent
the 6 sensors recording temperature and soil moisture content within the active layer. Red stars represent
the 4 sensors recording temperature and soil moisture content below the active layer. B) Ground
temperature distribution simulated by the ATS model, corresponding to the time of maximum active layer
depth. Here the depth of the active layer corresponds to the distance above the white linear feature (i.e.,
0°C) dividing the thawed and frozen regions of the ground. The light blue dots represent the location of
the electrodes in this setup.

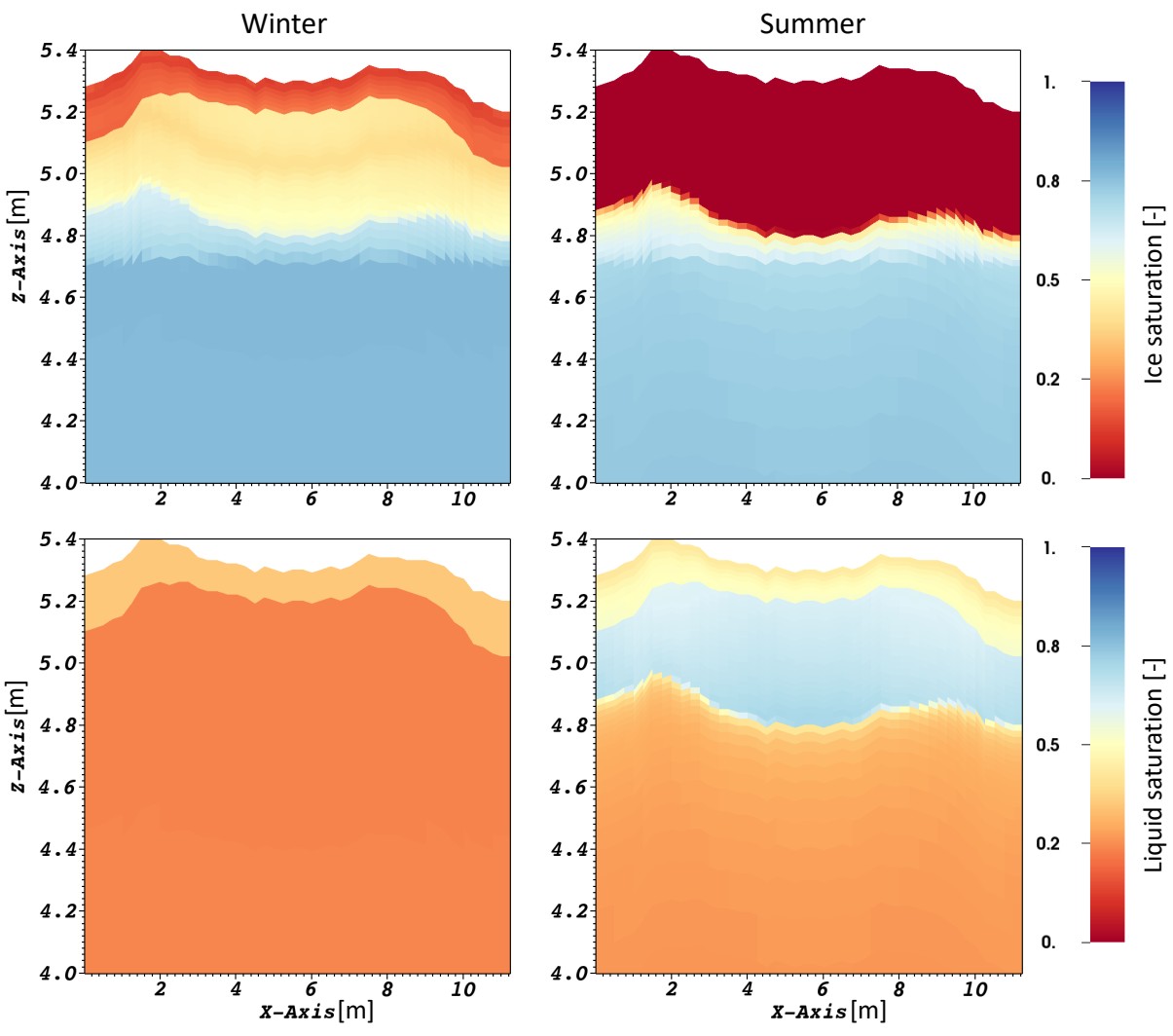

**Figure 3.** The 2D transect used by the ATS model. The rows from top to bottom correspond to ice saturation and liquid, respectively. The columns from left to right indicate one-day snapshots taken in the middle of the winter and the day of maximum active layer depth in summer.

## 2.2 Parameter estimation using PEST

To test if the known soil properties can be recovered by the PE approach, we start with randomly selected initial parameter guesses. We use a Latin Hypercube Sampling method to generate random initial guesses of porosity and thermal conductivities around the synthetic truth (McKay et al., 1979). Each parameter combination includes four parameters: porosity and thermal conductivity for peat and mineral soil layers. These parameters were chosen due to their strong controls on both hydrologic and thermal states (Atchley et al., 2015, Nicolsky et al., 2009). The rest of the hydrothermal properties are kept fixed.

The inverse approach involves the minimization of a cost function expressed as the sum-of-squared
differences between simulated values and synthetic measurements using the Levenberg-Marquardt
(LM) algorithm (K. Levenberg, 1944; D. W. Marquardt 1963) implemented in the PEST software
package (Doherty, 2001), which was used to handle all parameter estimation runs.
To estimate soil porosities and thermal conductivities, we minimize the cost function ($J$), which
includes calculated and synthetic $T$, $s_l$, and $\rho_a$ in the following form:
$$J(\phi, k) = w_T \sum_i^{n_{sens}} \sum_j^{n_{days}} \left(T_{cj}^i - T_{sj}^i\right)^2 + w_s \sum_i^{n_{sens}} \sum_j^{n_{days}} \left(s_{l\,cj}^i - s_{l\,sj}^i\right)^2 +$$

$$w_{\rho_a} \sum_k^{n_{snap}} \sum_m^{n_{meas}} \left(\rho_{a\,cm}^k - \rho_{a\,sm}^k\right)^2, \tag{5}$$

where subscripts $c$ and $s$ correspond to calculated and synthetic states of the system, and $w_T$, $w_s$,
and $w_{\rho_a}$ are the corresponding weights for the temperature, saturation and apparent resistivity
residuals. $n_{sens}$ is the number of sensors, $n_{days}$ is the number of days over which we collected the
data, $n_{snap}$ is the number of $\rho_a$ snapshots, and $n_{meas}$ is the number of $\rho_a$ measurements during
one snapshot. $T_c$ and $s_{l_c}$ are timeseries from multiple sensors collected daily from the beginning
of June till the end of September. $\rho_a$ are apparent resistivity data snapshots taken at a certain day.
The number of apparent resistivity snapshots depends on the particular case, varying from one to
eight snapshots per year.  The one-snapshot case corresponds to only one snapshot in the month
of August while the eight-snapshot case corresponds to a snapshot taken once per month from
January till September.  In addition, we tested the case where we collected eight daily $\rho_a$ snapshots.
This was done to compare how different time spacing would affect the estimated properties.
The weights were chosen in order to scale the contribution of each type of residual so that
contributions to the cost function are evenly distributed across temperature, saturation, and
apparent resistivity residuals. For example, saturation residuals are on the order of a few tenths,
while apparent resistivity residuals can be tens of ohm-meters. The weights were selected based
on evaluating the individual contributions to the cost function for each measurement type on an
ensemble of simulations spanning the parameter ranges. The apparent resistivity residual weight
($w_{\rho_a}$) was set to one. The temperature and saturation residual weights ($w_T$ and $w_s$) were then
modified so that each measurement type component in the cost function had roughly equivalent
magnitude over most of the parameter space. This resulted in weights of $w_{\rho_a} = 1$, $w_T =$
$\sqrt{2.5 \cdot 10^3}$, and $w_s = \sqrt{3.5 \cdot 10^5}$.
If the cost function satisfies a minimum criterion or the maximum allowed number of iterations,
which we chose to be equal to 25, is reached, the PE terminates. The porosities and thermal
conductivities corresponding to the minimum of the cost function, i.e., the parameters associated
with the best fit between simulated and synthetic values, are considered the estimated parameter
values as
$$\{\phi, k\} = \underset{\substack{\phi_{min} \leq \phi \leq \phi_{max}, \\ k_{min} \leq k \leq k_{max}}}{\mathrm{argmin}} J(\phi, k). \tag{6}$$

Here $\{\phi, k\}$, are estimated porosities and thermal conductivities for peat and mineral soil.
Based on sensitivity analyses using simplified meteorological data, the cost function response
surface was smooth and convex over the parameter ranges of interest. Therefore, we chose the LM
approach because of its robust gradient-based optimization scheme that takes advantage of smooth
convex response surfaces to quickly converge to minima.

**2.3 Experiments**
To build an understanding of the inverse framework, we start with a simple setup and then
gradually add more complexity.  First, we use simplified meteorological data where we assume
that air temperatures change according to a sinusoidal function and all other terms are constants.
Initially we start with 3 temperature and moisture content measurement locations within the peat
layer (refer to Figure 2A) and 1 ERT data snapshot.  Then we increase the number of ERT data
snapshots up to 8 by adding snapshots once per month from January till August. Each ERT data
snapshot calculated by BERT uses the set of daily averaged $T$ and $s_l$ simulated by ATS and
petrophysical relations (eqns. 2 and 3) which are varying over time.  Then we increase the number
of sensors to 6 and add noise to the simulated data.  Introduction of noise allows us to evaluate the
effect of measurement uncertainties that will be present in the actual application of the PE method.
We added different levels of Gaussian noise to the synthetic measurements of $T$, $s_l$, and $\rho_a$ in the
following way: 1% to $T$, 5% to $s_l$, and 10% to $\rho_a$. These levels of noise for the different types of
measurements are based on published literature and our own experience (Wang et al., 2018;
Dafflon et al., 2017). After that we substitute simplified meteorological data with actual data from
the BEO site to evaluate our PE method under realistic ground surface boundary conditions. In
this case we evaluate how much and what kind of data we need to robustly recover subsurface
porosities and thermal conductivities. To do this we test the inclusion of individual types of
measurements in the cost function (equation 3) as well as all possible combinations of
measurement types. We used different soil property ranges for the simplified and actual
meteorological data PE runs which are summarized in Table 1. This was done to ensure that PE is
able to recover different sets of parameters, and to test the consistency and effectiveness of the PE
method. Finally, we compared the difference between estimated parameters for 8 ERT data
snapshots collected once a month versus once a day for 8 days. Notation and a description of each
run for simplified and actual meteorological data are summarized in Table 2.
**Table 1: Allowed range for the estimated parameters.**

| Properties | Simplified meteorological data | | Actual meteorological data | |
|---|---|---|---|---|
| | **peat** | **mineral** | **peat** | **Mineral** |
| Porosity [$m^3 \cdot m^{-3}$] | 0.8±1.9 | 0.6+0.25 | 0.6±1.9 | 0.4+0.25 |
| Thermal conductivity, [$W\ m^{-1}K^{-1}$] | 0.225±0.2 | 2.0±0.5 | 0.15±0.1 | 1.6±0.5 |


**Table 2: Description of all PE cases used in this study**.
The numbers before $T$ and $s_l$ correspond to the number of sensors used. Number before $\rho_a$ corresponds
to the number of apparent resistivity snapshots used. $n$ stands for noise added to the synthetic
measurements. (S) corresponds to runs driven by simplified meteorological data. (s) represents daily $\rho_a$
snapshots.

| Case # | Simplified meteorological data (S) | | Actual meteorological data | |
|---|---|---|---|---|
| | **Case name** | **Description** | **Case name** | **Description** |
| 1 | (S)$3T3s_l1\rho_a$ | All data from #1 to #3 sensors, and 1 $\rho_a$ snapshot | 6T | Sensors from #1 to #6, temperature only |
| 2 | (S)$3T3s_l8\rho_a$ | All data from #1 to #3 sensors, and 8 $\rho_a$ snapshots | 10T | Sensors from #1 to #10, temperature only |
| 3 | (S)$6T6s_l1\rho_a$ | All data from #1 to #6 sensors, and 1 $\rho_a$ snapshot | $6s_l$ | Sensors from #1 to #6, liquid saturation only |
| 4 | (S)$6T6s_l8\rho_a$ | All data from #1 to #6 sensors, and 8 $\rho_a$ snapshots | $1\rho_a$ | 1 $\rho_a$ snapshot on month of August |
| 5 | (S)$6T6s_l1\rho_a+n$ | All data from #1 to #6 sensors, and 1 $\rho_a$ snapshot with added noise | $6T1\rho_a$ | Temperature sensors from #1 to #6, and 1 $\rho_a$ snapshots |
| 6 | (S)$6T6s_l8\rho_a+n$ | All data from #1 to #6 sensors, and 8 $\rho_a$ | $6s_l1\rho_a$ | Liquid saturation sensors from #1 to #6, and 1 $\rho_a$ snapshots |

| | | snapshots with added noise | | |
|---|---|---|---|---|
| 7 | | | $6T6s_l$ | Temperature and liquid saturation sensors from #1 to #6 |
| 8 | | | $3T3s_l1\rho_a$ | All data from #1 to #3 sensors, and 1 $\rho_a$ snapshot |
| 9 | | | $3T3s_l8\rho_a$ | All data from #1 to #3 sensors, and 8 $\rho_a$ snapshots |
| 10 | | | $6T6s_l8\rho_a$ | All data from #1 to #6 sensors, and 8 $\rho_a$ snapshots |
| 11 | | | $6T6s_l8\rho_a(s)$ | All data from #1 to #6 sensors, and 8 $\rho_a$ snapshots, taken every day |
| 12 | | | $6T6s_l1\rho_a$ | Special case, we moved sensors #4, #5 and #6 below the active layer depth (at 80cm depth), and 1 $\rho_a$ snapshot |
| 13 | | | $10T10s_l1\rho_a$ | All data from #1 to #10 sensors, and 1 $\rho_a$ snapshot |
| 14 | | | $10T10s_l8\rho_a$ | All data from #1 to #10 sensors, and 8 $\rho_a$ snapshots |


## 3. Results


### 3.1 Simplified meteorological data

To evaluate the PE method performance driven by simplified meteorological data, we ran PE
experiments using 30 different random combinations of porosity and thermal conductivity values
as the initial starting point. We used 30 PE samples of $\{\phi, k\}$ starting points in the first experiment
((S)$3T3s_l1\rho_a$) to illustrate the overall performance of the parameter estimation using a large
number of samples. After that, we did only five PE runs for the simplified meteorological data and
10 for all other runs with actual meteorological data. For all figures after Figure 4, for consistency
and clarity, we show results for only five PE runs per case. It is important to note that the number
of samples that one needs to run to ensure the robust convergence of the estimated parameters
depends on the specifics of the corresponding case (i.e. experiment specific). If most of the LM
runs converge to the same set of parameters and have low cost function values, then that set of
runs most likely corresponds to the actual $\{\phi, k\}$. In Figure 4, the red triangles represent initial
guesses (parameter combinations) and the synthetic truth is indicated by the intersection of the two
dotted lines. Yellow lines connecting red triangles with white crosses represent the path that the
LM algorithm has taken from the initial guess to the estimated parameter combination (white
crosses, Fig. 4). The yellow dots along the yellow lines indicate the location at each LM iteration.
Figure 4 indicates that the method is able to recover porosities more robustly than thermal
conductivities, i.e. estimated porosities are similar to their true state. According to the liquid
saturation plot on Figure 3, liquid saturation of the mineral layer is quite dynamic and more
saturated in comparison to the peat layer. Nevertheless, thermal conductivity of the mineral layer
($k_m$) corresponds to the highest mismatch. Three out of thirty inversions corresponding to $k_m$ end
up close to 1.4 $W\ m^{-1}K^{-1}$ (the true value is 2 $W\ m^{-1}K^{-1}$), suggesting those values do not
correspond to the 'truth', since most of the estimated values (27 cases) are concentrated around
the intersection of the dotted lines. The response surface for the corresponding cost function (eqn.
5) lies hereby in a flat, low-gradient region. The projections of the cost function response surfaces
corresponding to porosities (Figure 4A) has a better defined minimum, as opposed to projections
of the cost function response surfaces corresponding to thermal conductivities (Figure 4B),
indicating non-uniqueness of the estimated parameters. For this experiment, we used time-series
of $T$ and $s_l$ only from the first 3 near-surface sensors (Figure 2A). All of these 3 sensors are located
in the peat layer, suggesting that using just near-surface sensors only from one upper layer might
not be enough to recover the deeper layer thermal conductivity.

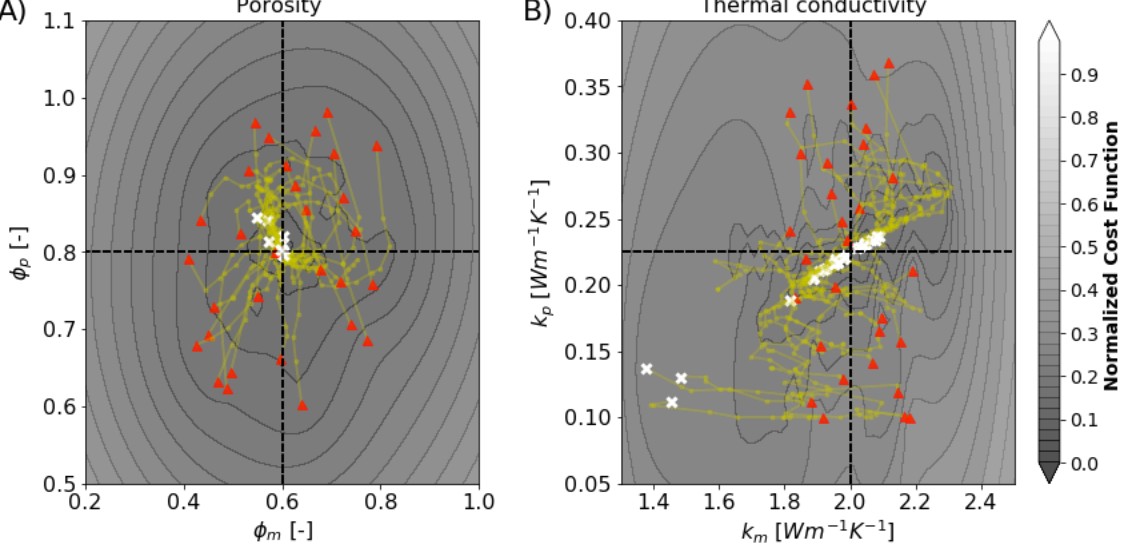


**Figure 4.** Estimated properties from 30 inversions of the (S)3T3$s_l$1$\rho_a$ case, where the "true" values are
shown as a cross-section of two dashed lines for the bulk porosities and effective thermal conductivities

364 for peat and mineral soil layers. Yellow lines correspond to the paths taken by the LM algorithm. The
365 white dots correspond to the estimated values. A) projection of the cost function with respect to porosities
366 of peat and mineral layer. B) projection of the cost function with respect to thermal conductivities of peat
367 and mineral layer. The color bar represents the cost function normalized by its maximum logarithmic
368 value.
369

370 To illustrate the effect of noise on the robustness of the estimated parameters we used cases with

371 6 near-surface sensors ($6T$ and $6s_l$), and a varying number of ERT snapshots driven with simplified

372 meteorological data. Similarly to the (S)$3T3s_l1\rho_a$ case, (S)$6T6s_l1\rho_a$ shows good convergence

373 for porosities and poor convergence for thermal conductivities with an averaged error of

374 $0.1Wm^{-1}K^{-1}$ (Figure 5AB). Adding noise to the (S)$6T6s_l1\rho_a+n$ case slightly worsens the

375 estimated porosity values and significantly worsens $k_m$ with root-mean-squared error (RMSE)

376 raising from 10% to more than 50% (Figure 5CD). Figure 5EF shows that increasing the number

377 of ERT snapshots from 1 to 8 per year (i.e. collected once per month from January till September)

378 improves $k_m$ estimates, allowing better convergence for four out of five samples to the synthetic

379 truth. If we compare all three cases on Figure 5 on how well they are able to estimate $k_m$, it is

380 clear from Figure 5D that for the case (S)$6T6s_l1\rho_a+n$ none of the $k_m$'s were correctly estimated,

381 whereas significantly improved $k_m$ values were found by increasing the number of monthly ERT

382 snapshots (Figure 5F). Moreover, all except one estimated value showed a better match with its

383 true value than the (S)$6T6s_l1\rho_a$ case without any added noise.

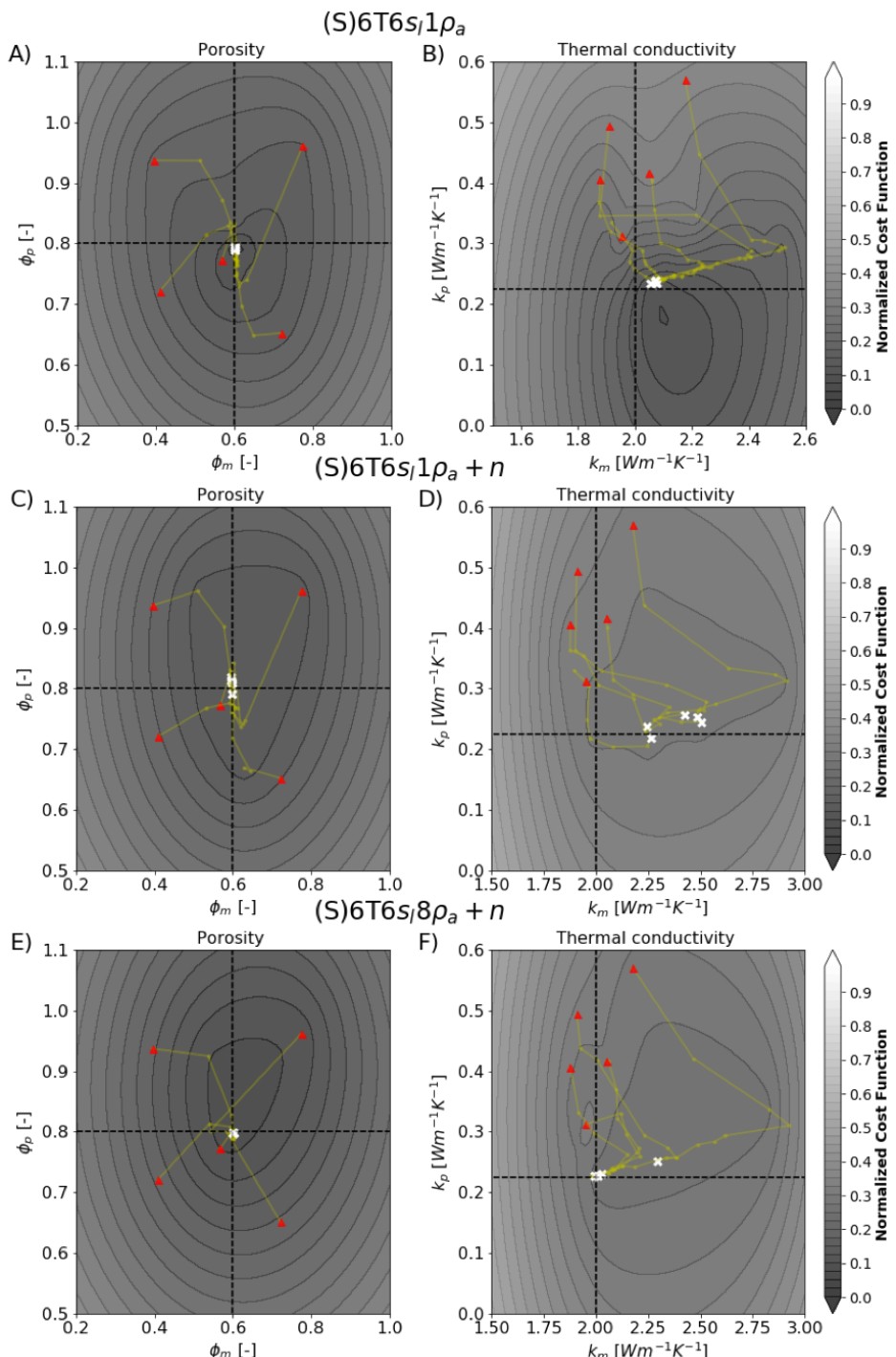

**Figure 5.** Estimated properties from 5 inversions of the three different cases, where the "true" values are shown as a cross-section of the two dashed lines for the bulk porosities and effective thermal conductivities for peat and mineral soil layers. Yellow lines correspond to the paths taken by the LM algorithm. The white dots correspond to the estimated values. The rows from top to bottom correspond to cases (S)6T6$s_l$1$\rho_a$, (S)6T6$s_l$1$\rho_a$+n, and (S)6T6$s_l$8$\rho_a$+n respectively. The columns from left to right correspond to the projection of the cost function with respect to porosities and thermal conductivities. The color bar represents the cost function normalized by its maximum logarithmic value.

In Figure 6, we summarize results of the five PE runs for each of the first six cases corresponding
to simplified meteorological data listed in Table 2. The first three matrix tables correspond to the
normalized RMSE values for each measurement type ($\Delta T$, $\Delta s_l$, and $\Delta \rho_a$). The last two matrix
tables correspond to the normalized Euclidian distances between the synthetic truth and estimated
parameter values of $\delta \phi$ and $\delta k$. We normalized the values in each matrix by dividing by the
maximum value from the corresponding matrix. The normalized values are marked with tildes
and range from 0 to 1, where values closer to 0 correspond to a better match and values closer to
1 correspond to a worse match. As shown above, the method is able to accurately estimate, both,
peat and mineral soil porosities as well as peat layer thermal conductivity ($k_p$), but cannot always
accurately estimate $k_m$. There is not much difference between cases (S)$3T3s_l1\rho_a$ and
(S)$6T6s_l1\rho_a$ except for a slight improvement in $k_m$, suggesting that the small vertical distance (10
cm) between sensors 1 and 4, 2 and 5, and 3 and 6 could be limit the recorded data variability,
leading to difficulties in the estimation of the $k_m$ parameter. Since all 6 sensors lie within the
active layer, we added additional sensors below the active layer in the later experiments (red stars
in Figure 2A). The $\phi$ and $k$ matrix tables show that increasing the number of monthly ERT
snapshots consistently improve the estimates of $\phi$ and $k$. This suggests that increasing the number
of monthly ERT snapshots can lead to improved convexity of the cost function (eqn.5).

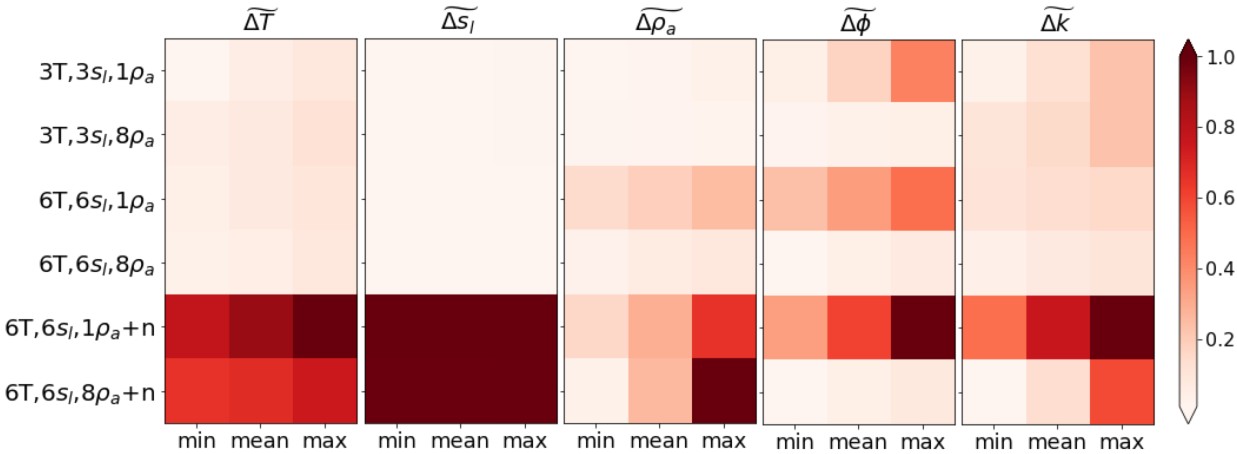


**Figure 6.** Five matrix tables presenting fitness metrics between synthetic model values and values
obtained by the parameter estimation method using simplified meteorological data. Matrix tables from left
to right correspond to the normalized root mean squared errors for temperatures, liquid water saturations,
and apparent resistivities and to the normalized Euclidian distances between synthetic ("true") and
estimated porosity, and thermal conductivity values. Each matrix value was normalized by dividing it by
the matrix maximum value. The normalized values are indicated by tildes.

418

## 3.2 Meteorological data from Utqiagvik (Barrow) site 2015

After testing the PE method for the simplified meteorological data, we applied measured meteorological data from the BEO site for the year 2015. To better understand the importance of each measurement type and their combinations within the developed PE algorithm, we tested all of the scenarios corresponding to the 'actual meteorological data' column from Table 2. The results of these runs are summarized in the colored matrix tables in Figure 7. Since there are more than twice the number of actual meteorological cases than simplified meteorological cases, it is difficult to analyze all matrix tables at once.

To compare the match between all estimated and observational values within a single plot we calculated Euclidean norms for each case independently:

$$\Delta\left(\widetilde{\Delta T}, \widetilde{\Delta s_l}, \widetilde{\Delta \rho_a}\right)_i = \sqrt{\left(\frac{\Delta T_i}{\Delta T_{max}}\right)^2 + \left(\frac{\Delta s_{l_i}}{\Delta s_{l_{max}}}\right)^2 + \left(\frac{\Delta \rho_{a_i}}{\Delta \rho_{max}}\right)^2} \qquad (7)$$

$$\Delta\left(\widetilde{\delta \phi}, \widetilde{\delta k}\right)_i = \sqrt{\left(\frac{\delta \phi_i}{\delta \phi_{max}}\right)^2 + \left(\frac{\delta k_i}{\delta k_{max}}\right)^2} \qquad (8)$$

The index $i$ indicates hereby the case number (see Table 2). Then we applied k-means clustering analysis to identify groups of cases with similar match between data and estimated parameters. We divided all cases into four classes shown in Figure 8. Class I indicates the best cases that provide an accurate parameter estimation as well as accurate matches with the synthetic "true" measurements. Class II includes the cases that have accurate parameter estimates and less accurate matches with the measurements. Class III indicates all cases that have less accurate parameter estimates but accurate matches with the measurements. Finally, Class IV includes the cases that showed the worst performance in terms of parameter estimates and the worst matches with the measurements. We summarized the results from Figure 8 in Table 3.

Class I (see Table 3) suggest that sensors located below the active layer as well as increasing the number of sensors lead to more accurate parameter estimation. In contrast, Case #4 (corresponding to the one ERT snapshot, $1\rho_a$), suggests that already one ERT data snapshot could be enough for parameter estimation while Class II indicates that in general an increase of the numbers of monthly ERT snapshots is important for more accurate PE. However, increasing the number of monthly ERT snapshots leads to a less accurate match with measurements. These results are consistent with the results for simplified meteorological data with added noise (Figure 5).

Class III includes 6 cases suggesting that if we have only soil moisture data available for PE, then we should expect less accurate soil property estimates. The last element in this class suggests that taking daily ERT snapshots improves the apparent resistivity match (Figure 7, resistivity table) but does not improve $\{\phi, k\}$ estimates, where monthly ERT snapshots improve thermal conductivity convergence.

Class IV once again clearly indicates that measurements obtained below the active layer provide more accurate parameter estimates, however, they do not improve matches to measurements. This is mainly due to significant mismatch with $\rho_a$, which can be seen from the $\widetilde{\Delta \rho_a}$ matrix table on Figure 8. At the actual site, the depth to the mineral soil can be deeper than 20 cm, not having sensors lower than 20 cm limits therefore the amount of data that can help to improve the convexity of the cost function in our case.

**Table 3:** K-mean analysis of the accuracy for each 13 cases.

| Class I | Class II | Class III | Class IV |
|---|---|---|---|
| $10T\,10s_l\,1\rho_a$ | $6T\,6s_l\,8\rho_a$ | $6s_l$ | $10T$ |
| $10T\,10s_l\,8\rho_a$ | $3T\,3s_l\,8\rho_a$ | $6s_l\,1\rho_a$ | $6T$ |
| $1\rho_a$ | | $6T\,1\rho_a$ | |
| $6T\,6s_l\,1\rho_a$ | | $3T\,3s_l\,1\rho_a$ | |
| | | $6T\,6s_l$ | |
| | | $6T\,6s_l\,8\rho_a(s)$ | |

From Figure 8 and Table 3 we know that the *6T* case has the worst performance in terms of matching $\{\phi, k\}$. Similar to the experiments with simplified meteorological data, the main difficulty for experiments with actual meteorological data is estimating thermal conductivity. The last matrix table ($\widetilde{\delta k}$) on Figure 7 shows that $6T\,6s_l\,8\rho_a(s)$ has the highest maximum and mean mismatch in thermal conductivity estimates. However, since $\phi$ estimates are a better match with their corresponding "true" values, the case $6T\,6s_l\,8\rho_a(s)$ falls into class III in Figure 8, as opposed to case *6T*, which falls into class IV. The highest mismatch in thermal conductivity values for the $6T\,6s_l\,8\rho_a(s)$ case suggests that collecting daily ERT snapshots improves the $\rho_a$ match (Figure 7, $\widetilde{\Delta \rho_a}$ matrix table) but does not improve estimated parameters, where monthly ERT snapshots improve thermal conductivity estimation.

To illustrate this, we plot values of estimated parameters and the corresponding response surfaces
of the cost function for cases $10T 10s_l 8\rho_a$ and $6T 6s_l 8\rho_a(s)$ on Figure 9. The PE method was able
to match 4 out of 5 estimates almost perfectly and missed the $k_p$ for the $10T 10s_l 8\rho_a$ case. The
corresponding cost function has a visible minimum and clear convexity. In contrast to this, the
$6T 6s_l 8\rho_a(s)$ case completely missed 2 estimates by converging on values outside the boundaries,
and 3 other estimates do not converge to the desired cross section as well. The contour lines suggest
that the corresponding response surfaces of the cost function do not have a well-defined global
minimum.

**4. Discussion**

The existence of multiple minima is common in inverse modeling and can lead to false
convergence of the PE algorithm to physically non-realistic subsurface parameters (Nicolsky et
al., 2007). This is one of the main reasons for using multiple initial guesses. If most of the
inversions converge to a similar set of parameter estimates with the lowest cost function value,
then that set of values is most likely the global minimum. Testing the PE algorithm using multiple
starting points is a commonly used approach in evaluating the robustness of an inverse model (e.g.
Hansen, 1998).
A potential strategy to improve the developed PE algorithms is to reduce the specified convergence
tolerance value (i.e. minimum condition, see Figure 1) or increase the allowable number of
iterations. However, this could lead to a significant increase in computational effort. In addition,
PEST provides multiple additional settings of inversion parameters to achieve a better
convergence. Parameter regularization is one of them. Regularization techniques have been
widely used in solving ill-posed inverse problems (Vogel, 2002). The overall idea is to constrain
the objective function by imposing additional priors on the estimated model parameters. We
recognize that including parameter regularization into the cost function may improve the
robustness of our method. However, inclusion of the regularization would require an extensive
exploration of the multiple regularization methods and values that could be applied to it, which is
beyond the scope of this paper. Here we illustrate that without using regularization it is possible to
achieve reasonable results by using the simple weighted cost function.
The good performance of the case with only one ERT snapshot ($1\rho_a$) could be misleading due to
the design of this numerical experiment, i.e. we are using a synthetic "truth" produced by the same
model used in the inversion, which improves the convexity of the cost function and leads to a well
constrained unique minimum. However, in reality, collection of additional information, such as
organic layer thickness and temperature data, are extremely important and are required for model
calibration (Jafarov et al., 2012; Atchley et al., 2015). In addition, real ERT surveys can be
perturbed by noise and their interpretation may require site-specific petrophysical relationships as
opposed to the general petrophysical relationships used in this study. Therefore, we do not suggest
inversion based on one ERT snapshot without any additional data.
The $6T6s_l8\rho_a(s)$ case, where all runs converge to different values of $k_m$, indicates that using
certain combinations of datasets does not allow the inverse approach to properly recover $k_m$. It is
likely that $6T6s_l8\rho_a(s)$ does not capture much variability in soil temperatures and soil moisture,
and therefore ERT snapshots do not have much variability as well. Once the cost function
converges for one of the ERT snapshots, it immediately converges on the other daily snapshots
due to their similarity. In fact, although $6T6s_l8\rho_a(s)$ has a good accuracy with observations (see
the $\widetilde{\Delta\rho_a}$ matrix table in Figure 7), it is unable to recover the value of $k_m$.
We have shown that even in the ideal situation where we either generate observational data or use
simplified meteorological data, we cannot always fit modeling results to observations. In reality,
noise (e.g. the sensor's measuring resolution) influences the collected data. To investigate the
impact of measurement noise, we introduced multiple levels of noise to the simplified
meteorological data. The resulting PE showed that dealing with noisy data could be challenging
(Figure 5). However, our results showed that adding monthly taken ERT snapshots into the cost
function improves the overall PE accuracy.
The distance between sensors could be another source of uncertainty in the PE. As pointed out by
Nicolsky et al., (2009), it is important to make sure that a vertical difference between adjacent
measurements do not introduce additional noise that can mislead the minimization algorithm
without providing new information. If sensors are close to each other, measurements might be the
same or within the noise variability. In our setup the vertical distance between the first two rows
of sensors is about 10cm. This could lead to small temperature variability between sensors. Indeed,
providing greater vertical distance between sensors improved the PE accuracy. The Case #12
($6T6s_l1\rho_a$) clearly illustrates this point, that by increasing the vertical distance between sensors
we can improve estimated parameter accuracy.
Combining hydrothermal observations from multiple depths with monthly ERT measurements
resulted in an improvement of the shape of the cost function and lead to better defined minima
(Figure 9). Increasing the number of the monthly ERT snapshots improved the accuracy of the
estimated parameters. In addition, we showed that having sensors below the active layer combined
with ERT snapshots shows the best accuracies, both, in terms of estimating parameters and
matching observations.

**5. Conclusion**
The overarching goal of this study was to develop and validate a parameter estimation algorithm
using a synthetic setup and a 2D coupled thermal-hydro-geophysical model based on a polygonal
tundra site within the Barrow Environmetal Observatory. Combining hydrothermal observations
from multiple depths with monthly ERT measurements resulted in an improved shape of the cost
function and led to better defined minima and improved accuracy of the estimated parameters.
This was presented in fitness matrices for six cases using simplified meteorological data. Similar
conclusion were found for inversion runs with actual meteorological data. It is important to note
that it was not only the number of ERT data snapshots that improved the robustness of the PE
method but rather the time frequency of the ERT data snapshots, i.e. monthly vs daily snapshots.
In addition, collecting data from several soil layers might improve the thermal conductivity
estimates for the corresponding soil layer. Our experiments show that robust PE can be achieved
not just by adding more sensors into the ground and increasing number of ERT snapshots, but also
by optimally distributing those sensors within the transect (e.g., the $6T6s_l1\rho_a$ case). Overall, the
inversion runs that we investigated consistently indicated that collecting data from multiple soils
layers, providing enough vertical separation between sensors, and collecting temporally diverse
ERT data should lead to robust parameter estimation. The exception from this conclusion is the
case $1\rho_a$, which showed robust parameter estimation due to specifics of the model setup. As
discussed above, estimating porosities and thermal conductivities based on 1 ERT snapshot would
not be possible without additional information on the subsurface properties.
This work developed and demonstrated the feasibility of a PE algorithm that can be used to better
inform large-scale Land System Model subsurface parameterization. Here we demonstrated the
proof-of-concept of the PE method. Further improvements such as introduction of a PE
regularization parameter into the cost function and leveraging additional PEST capabilities could
improve method robustness. Finally, the PE method must still be tested using measured thermal-
hydro-geophysical data from the BEO site.

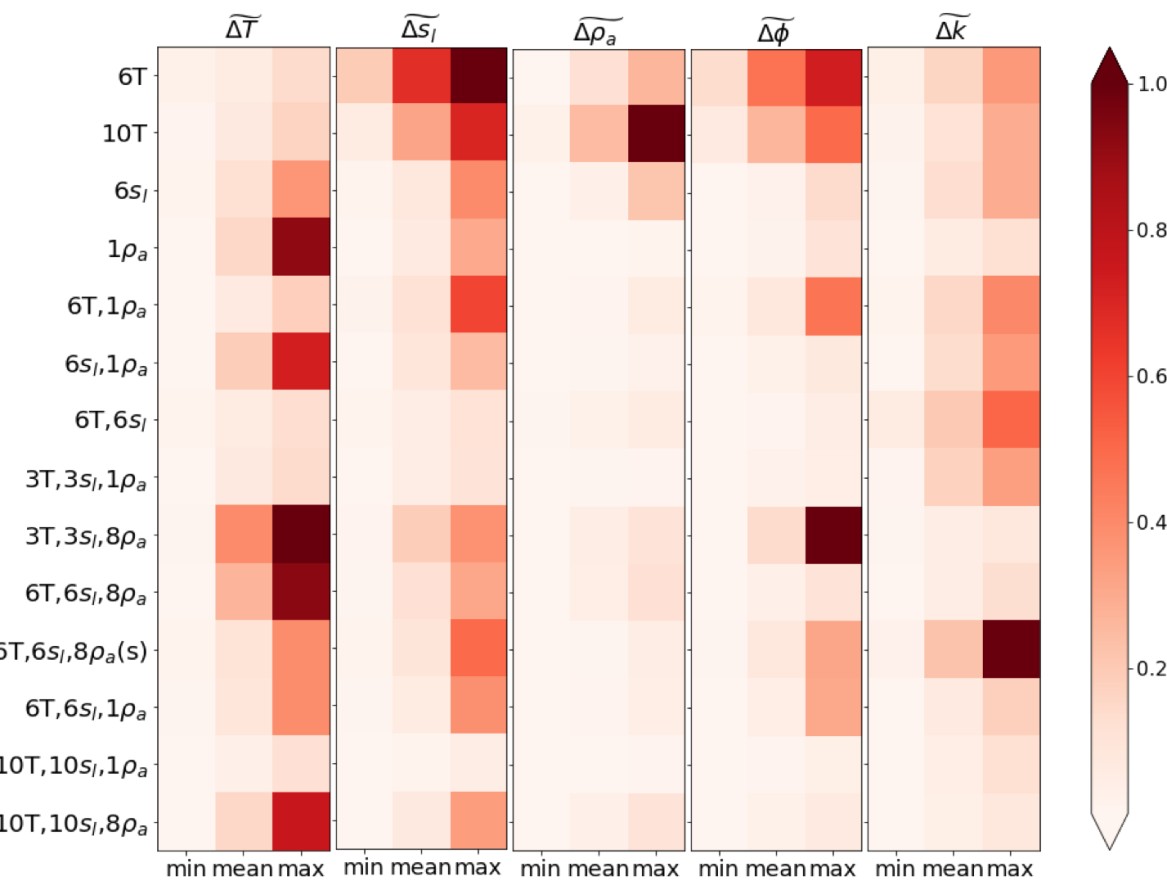


**Figure 7.** Five matrix tables presenting fitness metrics between synthetic model values and values
obtained by the parameter estimation method using meteorological data from the year 2015 from BEO
site in Alaska. Matrix tables from left to right correspond to the normalized root mean squared errors for
temperatures, liquid water saturations, and apparent resistivities, and to the normalized Euclidian
distances between synthetic ("true") and estimated porosity, and thermal conductivity values. Each matrix
value was normalized by dividing it by the matrix maximum value. The normalized values are indicated
by tildes.

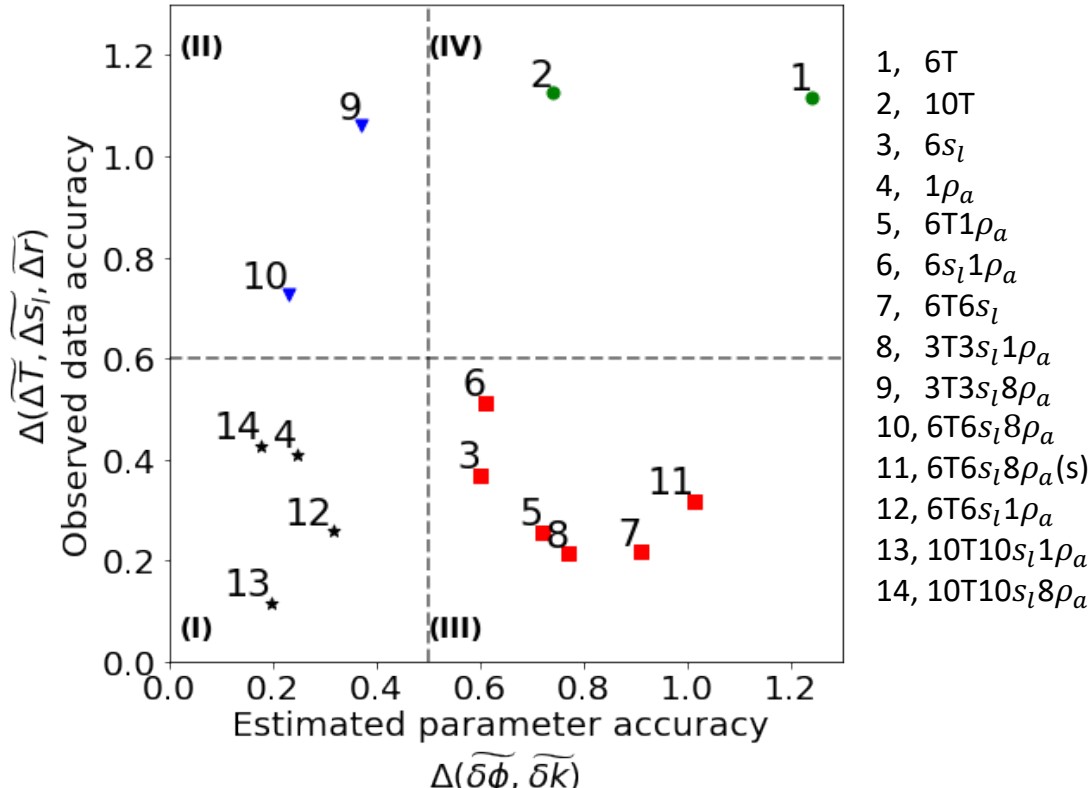


**Figure 8.** A k-means clustering analysis applied to the Euclidean norms of the normalized mean differences of estimated soil properties and the corresponding fit between calculated and observed values. Each color and marker represent a certain class as a result of the k-means clustering analysis.


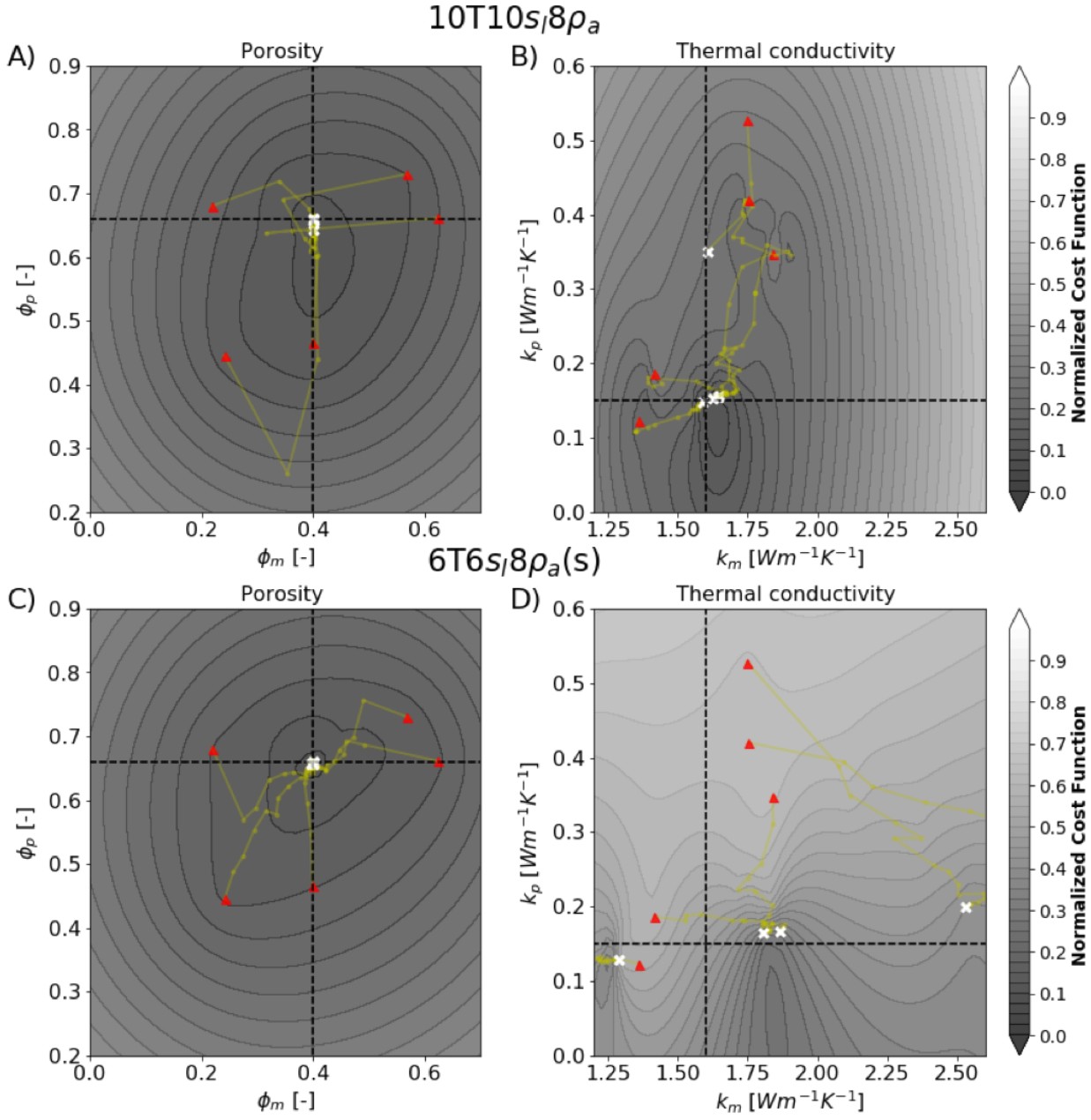

**Figure 9.** Estimated properties from five inversions of the two different cases: 10T10$s_l$8$\rho_a$ (top) and 6T6$s_l$8$\rho_a$(s) (bottom). The "true" values are shown as a cross-section of the two dashed lines for the bulk porosities and effective thermal conductivities for peat and mineral soil layers. Yellow lines correspond to the paths taken by the LM algorithm. The white dots correspond to the estimated values. The columns from left to right correspond to the projection of the cost function with respect to porosities and thermal conductivities. The color bar represents the cost function normalized by its maximum logarithmic value.

## 5. Acknowledgements

This work is part of the Next-Generation Ecosystem Experiments (NGEE Arctic) project which
is supported by the Office of Biological and Environmental Research in the DOE Office of
Science.

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
