# Peer review of "Estimation of subsurface porosities and thermal conductivities of polygonal tundra by coupled inversion of electrical resistivity, temperature, and moisture content data 3 Elchin E. Jafarov1, Dylan R. Harp1, Ethan T. Coon2, Baptiste Dafflon3, Anh Phuon"

_The Cryosphere, 2019_

## Referee Comment (RC1) · Anonymous Referee #1 · 7 Jun 2019

Summary

Developing a process-based understanding of permafrost thaw is an important research topic and in line with the scope of The Cryosphere (TC). The authors present an estimation of subsurface porosity and thermal conductivity during permafrost thaw based on a fully-coupled inversion of soil moisture, temperature, and electrical resistance data. While the first two quantities are provided by the physical model directly, the latter is achieved by linking temperature, liquid water saturation, and porosity to electrical resistivity for a subsequent process-based forward calculation. The parameter estimation framework is demonstrated on a synthetic case in analogy to a field

site near Barrow, Alaska. While this topic is definitely of interest for the readership of TC and should be considered for publication, the current manuscript has several flaws, which I detail in the general and more specific comments below. I hope that the authors take this feedback in the positive spirit intended during revision of this otherwise promising paper.

General comments

- Language: I feel that the writing in general should be more precise. For example, starting with the title, the authors refer to estimating "subsurface properties" several times. Porosity and thermal conductivity of peat and mineral layers are mentioned in line 48, but in a more general statement. Only in section 2 and Figure 1, the reader is finally introduced to the actual inversion parameters. These should be clearly stated much earlier (abstract, introduction, or even better, title). The unspecific term "subsurface parameters/properties" also claims generality where it is not appropriate, i.e. line 470: "The results of this study show that estimating subsurface properties even for a synthetic setup can be quite complicated.".

- Petrophysical coupling: This is still not clear to me. The authors state that they use the approach of Tran et al. (2017), where "subsurface temperature, liquid water content and ice content from CLM model were explicitly linked to soil electrical resistivities via petrophysical relationships", but there is no ice content in eq. 2. Instead, there is porosity, which in turn does not appear in Fig. 1 as an input to the third box. Porosities are estimated and hence updated during each iteration, are they not updated in eq. 2? Please clarify. The coupling is essential for the paper and should be clear to the reader without consulting additional literature.

- ERT layout and modeling: Essential information is missing in the manuscript: How many electrodes were used? What was the spacing? What type of measurement configuration (Dipole-Dipole, Wenner, Schlumberger, etc.) was used? How many measurements were made in total? Please clarify and consider adding the electrode locations to Fig. 2. Was the mesh in Fig. 2 also used for electrical forward modeling? How did the authors treat the fact that electrical fields diffuse much further than the parameter domain provided by the ATS simulations? Did the authors extrapolate to a larger mesh? The final recommendation on adding additional repeated ERT measurements strongly depends on the acquisition parameters, which are not discussed in the text.

- Objective function: The authors invert resistances. Why not follow the common practice and invert apparent resistivities instead? Resistances are not corrected for the geometry of the individual measurement, hence short-distance measurements (with higher sensitivity in the active layer) have less influence on the parameter estimation than measurements with larger layouts/geometric factors. The use of resistances could be justified with a more realistic error model, but w_r is set to 1 for all measurements, so the parameter estimation is dominated by measurements with a larger geometric factors, which would be more susceptible to noise in reality.

- Parameter estimation runs are performed with different starting models. While I think that this is a good idea, I do not agree that this is an appropriate measure of uncertainty, but rather of stability and suitability of the used inversion algorithm. If the Levenberg-Marquardt algorithm employed in this study gets trapped in local minima, one might raise the question if it is an appropriate approach? PEST provides some nice functionalities to study the sensitivity and resolving capacity of parameters. The manuscript could really benefit from a more detailed exploitation of sensitivities.

- Conclusions: The authors conclude that using more data as well as various types of observational data sets, and limiting the parameter ranges improves the parameter estimation. None of these conclusions are new to readers familiar with non-linear inverse problems. The conclusions should be more formulated more specifically to the present study and its implications for studying permafrost thaw.

Specific comments

- Fig 1: From the figure alone, one would draw the conclusion that only resistance

values are input to the objective function. Consider making a separate branch for the ERT part and adding an arrow from ATS (T, s) to the objective function.

- Fig. 7 & 8: The light colored numbers on top of the light colored matrix entries are not legible.

- Introducing (three) abbreviations already in the abstract is not necessary. In addition, there are some abbreviations (PCF, LSM), which are only used once and could thus be easily replaced by the full term.

- L26: Which soil properties?

- L30: Which specified subsurface parameters?

- L31: Which parameters?

- L34: Predictions of what?

- L64: Sorry for repeating myself, but "Nicolsky et al. (2007, 2009) used an optimization based inverse method [...] to estimate soil properties." is not very meaningful, i.e. the reader is left wondering: What type of observations were used? Which parameters were actually estimated? What was the outcome? How does it relate to the present study? I appreciate that additional information is given at the end of the paragraph, but in general too many statements in the text are too unspecific, which complicates reading.

- L66: Is "data calibration" different from "optimization based inverse method" and "parameter estimation"? The authors should strive to use a consistent wording.

- L93: Rephrase to "... matching multiple types of measurements to their model responses". Otherwise it is a bit misleading, as one could think that the different types of measurements are tried to be matched.

- L102-104: This is somewhat redundant to the statement in L99-101.

- L115: Replace by "etc." by something more specific. From equation 2, I assume that the porosity estimated in one iteration is also used in the petrophysical relation to calculate the corresponding electrical response. But this is not clear from Figure 1 and when reading the text.

- L228 and throughout the text: A "profile" is commonly used for a spatial location of measurements. Adding more profiles would certainly help, but I understand the authors take the exact same profile and propose more frequent measurements over time. Consider using a "measurements" instead of "profiles" to make this clear.

- L262: From what distribution were these random samples drawn? Did you also perform the parameter estimation with a homogeneous starting models, i.e. no differentiation between mineral and peat layer? This would be interesting.

- L273-274: Let the reader know how you came to this conclusion.

- Suggestion: I think that Figures 4 and 5 would be great in combination, e.g. the cost function as in Fig. 5 as an image (gray-scaled) in the background overlain by parameter trajectories on top as in Fig. 4. The same holds for Fig. 10 left and right.

- L296: Normalized to what? Maximum value? Please specify.

- L306: Why would one neglect the "outlier"? I thought the whole point of different starting models is an assessment of robustness? One out of five cannot be simply neglected.

- L417: "at least ten"? Why are there only five in Fig. 6 then? Please clarify.

- L424-426: I do not feel that the term "accuracy" is appropriate here. The "data accuracy" does certainly not depend on the regularization technique. Please choose a different wording.

- L426: "The regularization techniques". I suggest to leave out the "The" to be more general.

- References: Please use a consistent formatting of citations (e.g., sometimes the year appears at the end, sometimes after the authors, sometimes in brackets, doi: vs. DOI: vs. https://doi.org/).

Technical corrections

- L76: "soil electrical resistances" -> "soil electrical resistivities"

- L100: "total number measurements" -> "the total number of measurements"

- L189: Remove "as if the synthetic truth is unknown".

- L365: Remove "by"

- L396: "matching" -> "estimating" (to differentiate between data and model space)

- L427: "Its" -> "Their"

- L454: Replace "confuse"

- L462: "data" -> "parameters"

References

Tran, A. P., Dafflon, B., and Hubbard, S. S.: Coupled land surface–subsurface hydrogeophysical inverse modeling to estimate soil organic carbon content and explore associated hydrological and thermal dynamics in the Arctic tundra, The Cryosphere, 11, 2089-2109, https://doi.org/10.5194/tc-11-2089-2017, 2017.

---

## Referee Comment (RC2) · Anonymous Referee #2 · 9 Jul 2019

In the paper "Estimation of soil properties by coupled inversion of electrical resistance, temperature, and moisture content data" by Jafarov et al. the authors develop and explore a framework for estimation of soil properties (porosies and thermal conductivities) based on an optimization framework using Electrical Resistivity Tomography data and in-situ embedded sensor information. The paper is based on purely synthetic experiments, and aim to explore the types and frequency of data needed for good parameter estimation. While I find the topic of the paper very interesting and original, I also find that the paper has not been prepared with the attention to scientific quality and rigour that is to be expected from a paper published in The Cryosphere. I have outlined my concerns in the general and specific comments below, and hope that these comments

[Figure]

may help the authors prepare a revised version of the paper.

General comments:

The paper suffers somewhat from incosistent use of standard terms: Active Layer Thickness (ALT) is a well established term referring to the maximum thickness of the active layer (distance from surface to frost table), which is usually observed at the end of summer. The term is used inconsistently by the authors to refer to the "active layer" e.g. (L129), and the "frost table" (L147), and is used in several places to describe relative positions below the active layer ("below the ALT") where simply stating e.g. "below the active layer" or "in the permafrost" would be more clear to the reader. I would suggest that the authors implement the standard notation using the greek letter "rho" for resistivity and capital R for resistance to avoid confusion. (and that they use "rho_a" for apparent resistivity should they choose to change the representation of the ERT data in the cost function). For ERT measurements, the term "profile" is usually used to indicate the physical location of ERT measurements. The authors, however seem to use it to represent time slices of measurements along the same profile - which is confusing at first. It would help the reader if the authors implemented a more standard terminology using "profile" to refer to the physical location, and e.g. "data set" or "time slice" to refer to individual sets of measurements along the (single?) profile.

The petrophysical relationships used are essential for understanding the implementation and should be presented in the paper. One such relationship is given in equation 1, but no specific reference is provided for this relationship (although Tran 2017 is mentioned later as a general reference). It is unclear whether the soil is considered constantly fully saturated, or whether variation in saturation is modelled. It is unclear which component of thermal conductivity is optimized for (saturated frozen, saturated unfrozen, soil grain, ...?). $k_m$ and $k_p$ are mentioned (P15 L323-324) for mineral soil and peat. They are obviously single valued (no time variation), so what thermal conductivity component do they represent? What about thermal conductivity of water, ice and air - are they considered temperature invariant? What form of freezing characteristic curve has been chosen for the different soil types?

A critical problem of the paper is its faliure to describe the layout and measurement configurations of the ERT measurements. No mention is given to electrode locations and separation, and no discussion of measurement configurations is presented (4-electrode configurations I assume: wenner, gradient, schlumberger,...?). Furthermore, no discussion is presented of the sensitivity of the configurations used, and it is therefore unknown to the reader to which extent the layouts provide information within the domain of interest.

The data input to the optimization is not well described in terms of timeseries measurement frequency. Measurement frequency of temperature and liquid water saturation is not mentioned at all... are synthetic data used for a full year? 8 months like in some cases for the ERT? At what measurement frequency? ERT data sampling is confusingly described. It seems that the authors consider 3 situations in terms of ERT measurement frequency (P10 L227-229 and 243-245): A) One ERT data set (time of year is not specified) B) 8 ERT data sets, measured along the same profile, one data set per month for 8 months from January to August. C) 8 ERT datasets, measured along the same profile, one data set each day for 8 consecutive days (time of year is not specified) However, later a fourth case seem to be mentioned (P13 L304-305) with 8 monthly profiles (meaning 8 profiles collected every month? Equally spaced in time? Still for 8 months? so 64 data sets in total?). This must be much more clearly presented, and an explanation of the choice of timing (time of year) in the different cases must be given. This is very important as it will strongly influence the infomation contained in the resistivity data set.

It should be made more clear what data are used in the cost function calculations. I suggest the authors add and explain indices to the summations in the cost function, to represent both sensor locations / electrode configurations, and timing / time slices - to make it clear to the reader, how the cost function input is composed. For the temperature and liquid water content data, the positional index would be e.g. $i=1..10$

(representing the up to 10 locations indicated in figure 2) and a temporal index j=1...XX, would represent the number of observations included for each point. Similarly for the ERT data, the positional index would correspond to the individual 4-electrode measurement configurations used, i=1...YY, and the temporal index to the included data sets, e.g. j=1 for experiments with only one dataset, and j=1...8 for experiments with 8 data sets, etc.

The authors should explain why they chose to invert resistance values rather than apparent resistivities, as is standard practice. The measured resistances depend not only on the subsurface properties, but also strongly depend on the geometry of the electrode configuration used for the measurements. Furthermore, the authors have not chosen to linearize the problem, which is often done by applying a logarithmic transform to the ERT data. If this reflect active choices by the authors they should explain their considerations in the paper. If it does not, the authors should rethink their optimization strategy.

For the final analysis, the authors try to consolidate the the results of the RMSE tables (figure 8) using normalized measures to ease comparison. This is a good idea, however, I find the choice of normalization is unfortunate. The authors choose to normalize the mean RMSE by the maximum RMSE value obtained. The maximum RMSE corresponds to the worst fit obtained, and this value depend on the shape of the optimization surface, as well as the choice of intial values chosen by the operator for the parameter in a particular optimization experiment. The normalization basis is thus influenced by operator choice, and would be quite sensitive to outliers. I suggest the authors try to identify a different and more robust basis of noralization. I also do not understand how the authors calculate the "RMSE of the normalized mean values" (P17 L352), which are used in the clustering analysis. Please provide more detail or equations.

When these issues have been resolved, the discussion and conclusions should be reevaluated.

Minor and/or specific comments:

I suggest the authors complete their description of model initialization by also specifying how the model domain is initialized in terms of initial temperature distribution (only the lower boundary fixed temperature is currently presented) and initial water saturation of the domain. It would also benefit the reader to know if the authors have conducted some sort of spin-up, to ensure that the model domain is in equilibrium with the forcing data.

The authors define several abbreviations, some of which are used only few times. Please consider spelling out where abbreviation is not strictly necessary.

P5 Figure 1: Please expand the figure caption, and explain the parameters presented in the figure.

P5 L123: if this is a transect through an ice-wedge polygon site, why are the ice wedges not represented in the model?

L285-287: "The close proximity ...": very vague statement, please revise and elaborate. What is k_m?

P11 L262: If each experiment was repeated 30 times, why does figure 6 show only 5 samples? and how were these samples selected from the full set? How many repetitions were used for the cases forced by observed meteorological data?

P12/13 figure 4&5: Which experiment does these figures represent? Why do the axes extend to physically meaningless (negative) values? Why are the axes limits in figure 5 cropped to exclude some of the minima found in the optimization (around k=1.4 W/(m.K) for mineral soil). The contour interval chosen for figure 5 is insufficient to visualize the optimization path leading to the outlier solutions.

P14 figure 6: Plots should be labeled with the full experiment designation, (S)6T6sl1r, (S)6T6sl1r+n and (S)6T6sl8r+n. Again, the plot axes extend to physically meaningless values.

Figures 7 and 8: The text in the tables and table titles is not readable P17 L365-366: "... increasing the number of measurement locations, leads to more accurate parameter estimation". I don't believe you have actually documented wether it is the number of measurement points, or the choice of location of the measurement points that cause the improvement. What would happen if you still used 6 measurement locations, but distributed diferently (some deeper locations)?

P18 L389-391: "In reality, the depth of the mineral soil..." this statement is unclear, please revise.

P19 L397: misprint in matrix table reference, and wrong figure reference.

P21 L358-359: Again, you have not tested whether it is the location or number of observation points that matter.

P21 L475-476: "It is important to note that..." Unclear, please revise. Are you still referring to the ERT data?

P23 Figure 9: xlabel should read "Estimated properties accuracy" or similar.

P24 Figure 10: Axes extend to physically meaningless values.

---

## Author Comment (AC1) · 1 Oct 2019

We greatly thank our referees for their careful review and valuable comments which significantly strengthen our manuscript. We have modified the manuscript in response to these comments, and elaborate below on our responses (in dark blue) to their comments. Since some of the comments are similar, we incorporated both reviewers comments into one document. Each comment is numbered to ease navigation. First we reply to major comments then address editorial comments.

**Referee #1**

**Summary**

Developing a process-based understanding of permafrost thaw is an important research topic and in line with the scope of The Cryosphere (TC). The authors present an estimation of subsurface porosity and thermal conductivity during permafrost thaw based on a fully-coupled inversion of soil moisture, temperature, and electrical resistance data. While the first two quantities are provided by the physical model directly, the latter is achieved by linking temperature, liquid water saturation, and porosity to electrical resistivity for a subsequent process-based forward calculation. The parameter estimation framework is demonstrated on a synthetic case in analogy to a field site near Barrow, Alaska. While this topic is definitely of interest for the readership of TC and should be considered for publication, the current manuscript has several flaws, which I detail in the general and more specific comments below. I hope that the authors take this feedback in the positive spirit intended during revision of this otherwise promising paper.

**General comments**

**Comment #1**

- Language: I feel that the writing in general should be more precise. For example, starting with the title, the authors refer to estimating "subsurface properties" several times. Porosity and thermal conductivity of peat and mineral layers are mentioned in line 48, but in a more general statement. Only in section 2 and Figure 1, the reader is finally introduced to the actual inversion parameters. These should be clearly stated much earlier (abstract, introduction, or even better, title). The unspecific term "subsurface parameters/properties" also claims generality where it is not appropriate, i.e. line 470: "The results of this study show that estimating subsurface properties even for a synthetic setup can be quite complicated.".

We agree and changed "subsurface properties" in the manuscript to "subsurface porosities and thermal conductivities". In addition, we made the corresponding changes in the title. The new title reads: Estimation of subsurface porosities and thermal conductivities of polygonal tundra by coupled inversion of electrical resistivity, temperature, and moisture content data

In the introduction, we refer to the studies that estimated subsurface properties that are not necessarily porosity or thermal conductivity. To address this comment, we added the following clarification in the manuscript (L69-73):

"In particular, Nicolsky et al., (2007; 2009) used measured subsurface temperatures to inversely estimate thermal conductivities, porosities, freezing point temperatures, and unfrozen water

coefficients, pointing out that sensitivity analyses (i.e. perturbation of the parameter values) are required in order to robustly establish a set of estimated parameters."

**Comment #2**

- Petrophysical coupling: This is still not clear to me. The authors state that they use the approach of Tran et al. (2017), where "subsurface temperature, liquid water content and ice content from CLM model were explicitly linked to soil electrical resistivities via petrophysical relationships", but there is no ice content in eq. 2. Instead, there is porosity, which in turn does not appear in Fig. 1 as an input to the third box. Porosities are estimated and hence updated during each iteration, are they not updated in eq. 2? Please clarify. The coupling is essential for the paper and should be clear to the reader without consulting additional literature.

Correct, porosity and ice saturation are both changing with every new iteration, so we charged the notation and updated Figure 1 according to the comments. These properties are considered in the petrophysical relationship.

To clarify that we have modified the paragraph in the manuscript accordingly (L192-L212):

"We sequentially couple the ATS and BERT numerical models via petrophysical relationships used by Tran et al. (2017) and based on Archie (1942) and Minsley et al. (2015). In that approach, the electrical resistivity ( $\rho$ ) is determined as a function of soil characteristics, temperature, porosity, liquid water saturation, fluid conductivity, and ice content:

$$\rho = 1/(\phi^d [s_l^n \sigma_w + (\phi^{-d} - 1)\sigma_s] \cdot [1 + c(T - 25)]), \tag{2}$$

where  $\sigma_w$  is the fluid electrical conductivity,  $\sigma_s$  is soil/sediments electrical conduction, n is a saturation index, d is a cementation index, and c is a temperature compensation factor accounting for deviations from  $T = 25^{\circ}C$ .

The ice content is linked to water content through the liquid-water saturation and to  $\sigma_w$ , which is influenced by the concentration of Na+ and Cl- ions and the ice/liquid fraction. Here  $\sigma_w$  has the following form:

$$\sigma_{w} = \sum_{i=1}^{n_{ion}} F_{c} \beta_{i} |z_{i}| C_{i(S_{f_{i}=0})} S_{f_{w}}^{-\alpha}$$
(3)

Where  $F_c$  is Faraday's constant,  $\beta_i$  and  $z_i$  the ionic mobility and valence respectively,  $C_i$  is the concentration of  $i^{th}$  ion,  $\alpha$  is factor influencing how the liquid water salinity increases when the fractions of liquid in ice-liquid water  $S_{f_w}$  decreases.  $S_{f_w}$  is defined as:

$$S_{f_w} = s_l / (s_l + s_i) \tag{4}$$

Both  $s_l$  and  $s_i$  are simulated by ATS. Note that  $\phi$  in the eq. (2) is an estimated parameter (see Figure 1). In this study we assume that  $n, d, \sigma_s, \alpha, F_c, \beta_{Na^+}, \beta_{Cl^-}, C_{Na^+}$ , and  $C_{Cl^-}$  parameters used in equations (2) and (3) are known (see Tran et al., 2017) and focus on the robustness of the PE algorithm in estimating porosity and thermal conductivity."

**Comment #3**

ERT layout and modeling: Essential information is missing in the manuscript: How many electrodes were used? What was the spacing? What type of measurement configuration (Dipole-Dipole, Wenner, Schlumberger, etc.) was used? How many measurements were made in total?

Please clarify and consider adding the electrode locations to Fig. 2. Was the mesh in Fig. 2 also used for electrical forward modeling? How did the authors treat the fact that electrical fields diffuse much further than the parameter domain provided by the ATS simulations? Did the authors extrapolate to a larger mesh?

The final recommendation on adding additional repeated ERT measurements strongly depends on the acquisition parameters, which are not discussed in the text.

Figure 2B now shows spacing between the electrodes. To clarify the ERT configuration and different meshes used by ATS and BERT models we added the following paragraph into the manuscript (L212-225):

"The 2D resistivity data inferred from the ATS simulation and petrophysical relationships gets passed to BERT which simulates resistances that are then converted to  $\rho_a$ . The  $\rho_a$ values correspond to an acquisition along an 11 m long transect using a 0.5 m electrode spacing and a Schlumberger configuration with a total of 138 measurements (see Fig. 2B). This configuration implies that the measurements are mostly sensitive to the electrical resistivity in the top 2m.

Since BERT and ATS operate on different unstructured meshes, we wrote a function that interpolates the values between the two meshes. Note that ATS mesh is 50m deep.  $\rho$  is calculated by using corresponding outputs from ATS model and the petrophysical relations and then interpolated on a mesh defined in BERT and adapted to the acquisition geometry. BERT's mesh consists of a finely resolved mesh (11m long by 4.5m deep) and coarser outer mesh that is about 120m long and 85m deep. We link hydrological variables with electrical resistivities in the fine mesh. The coarse mesh is used to reduce the effect of boundary. It extends until the change in the electrical resistivity between two neighbors cells is negligible."

**Comment #4**

- Objective function: The authors invert resistances. Why not follow the common practice and invert apparent resistivities instead? Resistances are not corrected for the geometry of the individual measurement, hence short-distance measurements (with higher sensitivity in the active layer) have less influence on the parameter estimation than measurements with larger layouts/geometric factors. The use of resistances could be justified with a more realistic error model, but w\_r is set to 1 for all measurements, so the parameter estimation is dominated by measurements with a larger geometric factors, which would be more susceptible to noise in reality.

We agree and thank the reviewer for indicating this. This was simply an error in terminology on our part. We do correct for the geometry, so the compared final values are, in fact, apparent resistivities. We made the corresponding changes in the cost function (L255-267).

$$"J(\phi, k) = w_T \sum_{i}^{n_{sens}} \sum_{j}^{n_{days}} (T_{cj}^i - T_{sj}^i)^2 + w_s \sum_{i}^{n_{sens}} \sum_{j}^{n_{days}} (s_{l_{cj}}^i - s_{l_{sj}}^i)^2 + w_{\rho_a} \sum_{k}^{n_{snap}} \sum_{m}^{n_{meas}} (\rho_{a_{cm}}^k - \rho_{a_{sm}}^k)^2,$$
(5)

where subscripts c and s correspond to calculated and synthetic states of the system,  $w_T$ ,  $w_s$ , and  $w_{\rho_a}$  are the corresponding weights for the temperature, saturation and apparent resistivity residuals.  $n_{sens}$  is the number of sensors,  $n_{days}$  is the number of days over which we collected the data,  $n_{snap}$  is the number of  $\rho_a$  snapshots, and  $n_{meas}$  is the number of  $\rho_a$  measurements during one snapshot.  $T_c$  and  $s_{l_c}$  are timeseries from multiple sensors collected daily from the beginning of June till the end of September.  $\rho_a$  are apparent resistivity snapshots depends on the particular case, varying from one to eight snapshots per year. The one-snapshot case corresponds to only one snapshot in the month of August while the eight-snapshot case corresponds to a snapshot taken once per month from January till September. In addition, we tested the case where we collected eight daily  $\rho_a$  snapshots. This was done to compare how different time spacing would affect the estimated properties."

**Comment #5**

- Parameter estimation runs are performed with different starting models. While I think that this is a good idea, I do not agree that this is an appropriate measure of uncertainty, but rather of stability and suitability of the used inversion algorithm. If the Levenberg-Marquardt algorithm employed in this study gets trapped in local minima, one might raise the question if it is an appropriate approach? PEST provides some nice functionalities to study the sensitivity and resolving capacity of parameters. The manuscript could really benefit from a more detailed exploitation of sensitivities.

We agree, we should not be calling the match between true and estimated values uncertainty. We changed that to robustness to capture the reviewer's correct interpretation that this is really indicating the "stability and suitability of the used inversion algorithm".

We added the following paragraph into the Discussion section (L472-490):

"The existence of multiple minima is common in inverse modeling and can lead to false convergence of the PE algorithm to physically non-realistic subsurface parameters (Nicolsky et al., 2007). This is one of the main reasons for using multiple initial guesses. If most of the inversions converge to a similar set of parameter estimates with the lowest cost function value, then that set of values is most likely the global minimum. Testing the PE algorithm using multiple starting points is a commonly used approach in evaluating the robustness of an inverse model (e.g. P. Hansen, 1998).

A potential strategy to improve of the developed PE algorithms is to reduce the specified convergence tolerance value (i.e. minimum condition, see Figure 1) or increase the allowable number of iterations. However, this could lead to a significant increase in computational effort. In addition, PEST provides multiple additional inverse settings to

achieve a better convergence. Parameter regularization method is one of them. Regularization techniques have been widely used in solving ill-posed inverse problems (Vogel, 2002). The overall idea is to constrain the objective function by imposing additional priors on the estimated model parameters. We recognize that including parameter regularization into the cost function may improve robustness of our method. However, inclusion of the regularization would require an extensive exploration of the multiple regularization methods and values that could be applied to it, which is beyond the scope of this paper. Here we illustrate that without using regularization it is possible to achieve plausible results by using simple weighted cost function."

**Comment #6**

Conclusions: The authors conclude that using more data as well as various types of observational data sets, and limiting the parameter ranges improves the parameter estimation. None of these conclusions are new to readers familiar with non-linear inverse problems. The conclusions should be more formulated more specifically to the present study and its implications for studying permafrost thaw.

We see the reviewer's point and have completely rewritten the conclusion within the context of the present study focusing on permafrost thaw(L531-555):

"The overarching goal of this study was to develop and validate a PE algorithm using a synthetic setup and 2D coupled thermal-hydro-geophysical model based on a polygonal tundra site within the BEO. Combining hydrothermal observations from multiple depths with monthly ERT measurements resulted in an improved shape of the cost function and led to better defined minima and improved accuracy of the estimated parameters. This was presented in fitness matrices for 6 cases using simplified meteorological data (Figure 6). Similar conclusions held for inversion runs with actual meteorological data. It is important to note that it was not only the number of ERT data snapshots that improved the robustness of the PE method but time frequency of the ERT data snapshots, i.e. monthly vs daily snapshots. In addition, collecting data from several soil layers might improve the thermal conductivity estimates for the corresponding soil layer. Our experiments show that robust PE can be achieved not just by adding more sensors into the ground and increasing number of ERT snapshots, but also by optimally distributing those sensors within the transect (e.g., the  $6T6s_1 1\rho_a$  case). Overall, the inversion runs that we investigated consistently indicated that collecting data from multiple soils layers, providing enough vertical separation between sensors, and collecting temporally diverse ERT data should lead to robust parameter estimation. The exception from this conclusion is the case  $I\rho_a$ , which showed robust parameter estimation due to specifics of the model setup. As discussed above, estimating porosities and thermal conductivities based on 1 ERT snapshot would not be possible without additional information on the subsurface properties.

This work developed and demonstrated the feasibility of a PE algorithm that can be used to better inform large scale Land System Model subsurface parameterization. Here we demonstrated the proof-of-concept of the PE method. Further improvements such as introduction of a PE regularization parameter into the cost function and leveraging additional PEST capabilities could improve method robustness. Finally, the PE method must still be tested using measured thermal-hydro-geophysical data from the BEO site."

**Referee #2**

In the paper "Estimation of soil properties by coupled inversion of electrical resistance, temperature, and moisture content data" by Jafarov et al. the authors develop and explore a framework for estimation of soil properties (porosies and thermal conductivities) based on an optimization framework using Electrical Resistivity Tomography data and in-situ embedded sensor information. The paper is based on purely synthetic experiments, and aim to explore the types and frequency of data needed for good parameter estimation. While I find the topic of the paper very interesting and original, I also find that the paper has not been prepared with the attention to scientific quality and rigour that is to be expected from a paper published in The Cryosphere. I have outlined my concerns in the general and specific comments below, and hope that these comments may help the authors prepare a revised version of the paper.

General comments:

**Comment #7**

The paper suffers somewhat from incosistent use of standard terms: Active Layer Thickness (ALT) is a well established term referring to the maximum thickness of the active layer (distance from surface to frost table), which is usually observed at the end of summer. The term is used inconsistently by the authors to refer to the "active layer" e.g. (L129), and the "frost table" (L147), and is used in several places to describe relative positions below the active layer ("below the ALT") where simply stating e.g. "below the active layer" or "in the permafrost" would be more clear to the reader.

We agree with the reviewer and made the corresponding changes throughout the text. (L145-148)

"We initially designated six synthetic observation (temperature and soil moisture measurement) locations within the active layer area, the maximum thaw layer from ground surface to permafrost, similar to the sensor setup at the site (Dafflon et al., 2017)."

**Comment #8**

I would suggest that the authors implement the standard notation using the greek letter "rho" for resistivity and capital R for resistance to avoid confusion. (and that they use "rho\_a" for apparent resistivity should they choose to change the representation of the ERT data in the cost function). For ERT measurements, the term "profile" is usually used to indicate the physical location of ERT measurements. The authors, however seem to use it to represent time slices of measurements along the same profile - which is confusing at first. It would help the reader if the authors implemented a more standard terminology using "profile" to refer to the physical

location, and e.g. "data set" or "time slice" to refer to individual sets of measurements along the (single?) profile.

We made revisions throughout the manuscript according to the reviewer's suggestion. The current version of the manuscript has corrected notation for resistivity  $\rho$  and apparent resistivity  $\rho_a$  correspondingly.

We agree that "profile" mainly describes the vertical profile of a property. According to the reviewer's suggestion, we changed the term "profile" to "snapshot" everywhere in the text.

**Comment #9**

The petrophysical relationships used are essential for understanding the implementation and should be presented in the paper. One such relationship is given in equation 1, but no specific reference is provided for this relationship (although Tran 2017 is mentioned later as a general reference). It is unclear whether the soil is considered constantly fully saturated, or whether variation in saturation is modelled. It is unclear which component of thermal conductivity is optimized for (saturated frozen, saturated unfrozen, soil grain, ...?). k\_m and k\_p are mentioned (P15 L323-324) for mineral soil and peat. They are obviously single valued (no time variation), so what thermal conductivity component do they represent? What about thermal conductivity of water, ice and air - are they considered temperature invariant? What form of freezing characteristic curve has been chosen for the different soil types?

We added more details on how saturation liquid and saturation ice have been used in petrophysical relations. Please refer to the petrophysical relations comment #2.

We appreciate the reviewer pointing out that some aspects of the process descriptions included in ATS were not clear, and have made adjustments to the text to clarify(L173-186):

"Critical for these simulations is the calculation of the thermal conductivities of the bulk soil; calculated in ATS using Kersten numbers to interpolate between saturated frozen, saturated unfrozen, and fully dry states (Painter et al., 2016) where the thermal conductivities of each end-member state is determined by the thermal conductivity of the components (soil grains, air, water, or liquid) weighted by the relative abundance of each component in the cell (Johansen, 1977; Peters-Lidard et al, 1998; Atchley et al., 2015). Thermal conductivities of water, ice, and air are considered constant, leaving soil grain thermal conductivity as the remaining parameter to be estimated. The corresponding equation used to calculate saturated, frozen thermal conductivity ( $\kappa_{sat,f}$ ) has the following form:

$$\kappa_{sat,f} = \kappa_{sat,uf} \cdot \kappa_i^{\phi} \cdot \kappa_w^{-\phi}, \tag{1}$$

where  $\kappa_{sat,uf}$ ,  $\kappa_i$ ,  $\kappa_w$  are thermal conductivities for saturated unfrozen, ice, and liquid water, respectively, and  $\phi$  is porosity.

The freezing characteristic curve is thermodynamically derived using a Clapeyron relation and the unfrozen water retention curve, as described in Painter and Karra (2014) and Painter et al., (2016)."

**Comment #10**

A critical problem of the paper is its faliure to describe the layout and measurement configurations of the ERT measurements. No mention is given to electrode locations and separation, and no discussion of measurement configurations is presented (4- electrode configurations I assume: wenner, gradient, schlumberger,...?). Furthermore, no discussion is presented of the sensitivity of the configurations used, and it is therefore unknown to the reader to which extent the layouts provide information within the domain of interest.

We agree and addressed this in a similar comment from Referee #1 above. We also added electrode location to the Fig 2B. Please see Comment #3.

**Comment #11**

The data input to the optimization is not well described in terms of timeseries measurement frequency. Measurement frequency of temperature and liquid water saturation is not mentioned at all... are synthetic data used for a full year? 8 months like in some cases for the ERT? At what measurement frequency? ERT data sampling is confusingly described. It seems that the authors consider 3 situations in terms of ERT measurement frequency (P10 L227-229 and 243-245): A) One ERT data set (time of year is not specified) B) 8 ERT data sets, measured along the same profile, one data set per month for 8 months from January to August. C) 8 ERT datasets, measured along the same profile, one data set each day for 8 consecutive days (time of year is not specified) However, later a fourth case seem to be mentioned (P13 L304-305) with 8 monthly profiles (meaning 8 profiles collected every month? Equally spaced in time? Still for 8 months? so 64 data sets in total?). This must be much more clearly presented, and an explanation of the choice of timing (time of year) in the different cases must be given. This is very important as it will strongly influence the infomation contained in the resistivity data set.

We added clarification on the timeseries parameters in the cost function in the text. Please refer to Comment #4 for details.

However, later a fourth case seem to be mentioned (P13 L304-305) with 8 monthly profiles (meaning 8 profiles collected every month? Equally spaced in time? Still for 8 months?

We addressed this issue and apologize for the confusion (L367-370).

"Figure 5EF shows that increasing the number of ERT snapshots from 1 to 8 per year (i.e. collected once per month from January till September) improves  $k_m$  estimates, allowing better convergence for four out of five samples to the synthetic truth."

Comment #12

It should be made more clear what data are used in the cost function calculations. I suggest the authors add and explain indices to the summations in the cost function, to represent both sensor locations / electrode configurations, and timing / time slices - to make it clear to the reader, how the cost function input is composed. For the temperature and liquid water content data, the positional index would be e.g. i=1..10 (representing the up to 10 locations indicated in figure 2) and a temporal index j=1...XX, would represent the number of observations included for each point. Similarly for the ERT data, the positional index would correspond to the individual 4-electrode measurement configurations used, i=1...YY, and the temporal index to the included data sets, e.g. j=1 for experiments with only one dataset, and j=1...8 for experiments with 8 data sets, etc.

We agree and implemented the reviewer's suggested clarifications. Please refer to Comment #3 and updated Figure 2.

**Comment #13**

The authors should explain why they chose to invert resistance values rather than apparent resistivities, as is standard practice. The measured resistances depend not only on the subsurface properties, but also strongly depend on the geometry of the electrode configuration used for the measurements. Furthermore, the authors have not chosen to linearize the problem, which is often done by applying a logarithmic transform to the ERT data. If this reflect active choices by the authors they should explain their considerations in the paper. If it does not, the authors should rethink their optimization strategy.

While we erroneously referred to the calibration targets as "resistances", we actually did invert for "apparent resistivities". We corrected the previous notation in the text to "apparent resistivities". Please see Comment #4.

**Comment #14**

For the final analysis, the authors try to consolidate the the results of the RMSE tables (figure 8) using normalized measures to ease comparison. This is a good idea, however, I find the choice of normalization is unfortunate. The authors choose to normalize the mean RMSE by the maximum RMSE value obtained. The maximum RMSE corresponds to the worst fit obtained, and this value depend on the shape of the optimization surface, as well as the choice of intial values chosen by the operator for the parameter in a particular optimization experiment. The normalization basis is thus influenced by operator choice, and would be quite sensitive to outliers. I suggest the authors try to identify a different and more robust basis of noralization.

I also do not understand how the authors calculate the "RMSE of the normalized mean values" (P17 L352), which are used in the clustering analysis. Please provide more detail or equations.

We chose the RMSE norm because it is most commonly used and most obvious comparison measure. We also tried absolute difference and weighted norm, none of those measures changed the coloring in the matrix tables. In our case we know the 'true' values for all parameters from the cost functions so the square root of mean square measure works the best. We agree the way it was written was in fact confusing. We change that to the following (418-422):

"To compare the match between estimated and observational values on the same plot we calculated Euclidean norms for each case independently:

$$\Delta \left( \widetilde{\Delta T}, \widetilde{\Delta s_{l}}, \widetilde{\Delta \rho_{a}} \right)_{i} = \sqrt{\left( \frac{\Delta T_{i}}{\Delta T_{max}} \right)^{2} + \left( \frac{\Delta s_{l_{i}}}{\Delta s_{l_{max}}} \right)^{2} + \left( \frac{\Delta \rho_{a_{i}}}{\Delta \rho_{max}} \right)^{2}}$$
$$\Delta \left( \widetilde{\delta \phi}, \widetilde{\delta k} \right)_{i} = \sqrt{\left( \frac{\delta \phi_{i}}{\delta \phi_{max}} \right)^{2} + \left( \frac{\delta k_{i}}{\delta k_{max}} \right)^{2}}$$

The index *i* indicates the case number (see Table 1)."

In addition, we redid both Figures showing mismatch between measured and estimated data, as well as added the following clarification in the Results section:

"We normalized the values in each matrix by dividing by the maximum value from the corresponding matrix. The normalized values are marked with tildes and range from 0 to 1, where values closer to 0 correspond to a better match and values closer to 1 correspond to a worse match."

When these issues have been resolved, the discussion and conclusions should be reevaluated.

We completed all the major comments and below address minor comments.

Minor and/or specific comments:

**Reviewer #1**

- Fig 1: From the figure alone, one would draw the conclusion that only resistance values are input to the objective function. Consider making a separate branch for the ERT part and adding an arrow from ATS (T, s) to the objective function.

We made the corresponding changes in the Figure 1 (see comment #2).

- Fig. 7 & 8: The light colored numbers on top of the light colored matrix entries are not legible.

We got rid of the number, since those numbers are not included in the discussion, and the color range more clearly indicates the misfits now. New figures have color matrices only. We also normalized them for clarity and consistency in the manuscript.

- Introducing (three) abbreviations already in the abstract is not necessary. In addition, there are some abbreviations (PCF, LSM), which are only used once and could thus be easily replaced by the full term.

We removed the abbreviations "PCF" and "LSM" from abstract and introduction.

- L26: Which soil properties?

- L30: Which specified subsurface parameters? - L31: Which parameters?

These two comments were addressed in Comment #1. Current version of the manuscript explicitly states "subsurface porosities and thermal conductivities" everywhere throughout of the manuscript.

**- L34: Predictions of what?**

We removed that sentence and rewrote the ending of the Abstract to the following:

"In addition, we varied types and quantities of data to better understand the optimal dataset needed to improve the PE method. The results of the PE tests suggest that using multiple types of data improve the overall robustness of the method. Our numerical experiments show that special care needs to be taken during data collection and the experiment setup, so that vertical distance between adjacent temperature and soil moisture measuring sensors allows the signal variability in space and longer time interval between resistivity snapshots allows signal variability in time."

- L64: Sorry for repeating myself, but "Nicolsky et al. (2007, 2009) used an optimization based inverse method [...] to estimate soil properties." is not very meaningful, i.e. the reader is left wondering: What type of observations were used? Which parameters were actually estimated? What was the outcome? How does it relate to the present study? I appreciate that additional information is given at the end of the paragraph, but in general too many statements in the text are too unspecific, which complicates reading.

We agree and addressed this comment in more details in our reply to the first major comment (Comment #1).

- L66: Is "data calibration" different from "optimization based inverse method" and "parameter estimation"? The authors should strive to use a consistent wording.

Yes, the actual methodologies applied by different researchers to estimate desired parameters vary. We made sure that we use consistent wording for the rest of the manuscript. However, we cannot call by one-word different methods implemented by different researchers. In addition we would like to be consistent with the terminology that researchers used in the studies.

- L93: Rephrase to "... matching multiple types of measurements to their model responses". Otherwise it is a bit misleading, as one could think that the different types of measurements are tried to be matched.

We did, thank you for catching this. We modified the entire sentence to (L250-253):

"The inverse modeling framework couples the state-of-the-art hydrothermal permafrost simulator ATS, electrical resistivity software package BERT and the Model Independent Parameter Estimation and Uncertainty Analysis toolbox (PEST) software package (Doherty, 2001)."

- L102-104: This is somewhat redundant to the statement in L99-101.

We agree. We removed the repeated statement and rewrote the ending of the Introduction Section in the following way (L110-114):

"The results of this work can be used to better understand challenges associated with subsurface porosity and thermal conductivity estimation. Additionally, we used findings from this study to suggest how data should be collected to improve the accuracy of the estimated soil properties and to optimize the total number of measurements needed to make a robust subsurface PE."

- L115: Replace by "etc." by something more specific. From equation 2, I assume that the porosity estimated in one iteration is also used in the petrophysical relation to calculate the corresponding electrical response. But this is not clear from Figure 1 and when reading the text.

For more clarity, we rephrase the entire sentence (124-127):

"Given this hydrothermal state, we infer resistivity values at every grid cell via petrophysical relationships, and run the forward modelling component of the BERT software package (Rücker et al., 2006) to simulate resistance and related apparent resistivity values that would be measured with ground-coupled electrodes and an ERT acquisition system."

- L228 and throughout the text: A "profile" is commonly used for a spatial location of measurements. Adding more profiles would certainly help, but I understand the authors take the exact same profile and propose more frequent measurements over time. Consider using a "measurements" instead of "profiles" to make this clear.

We agree. We changed "profile" to "snapshot", which better describes the 2D resistivity measurement. This issue is addressed in Comment #8 as well.

- L262: From what distribution were these random samples drawn? Did you also perform the parameter estimation with a homogeneous starting models, i.e. no differentiation between mineral and peat layer? This would be interesting.

Yes, the samples are randomly drawn using Latin Hypercube Sampling method which allows to fill the entire space with non-repeating number of samples which optimally cover the parameter space. This is described in the section 2.2 Parameter estimation (PEST). In this case our parameter space consists of porosities and thermal conductivities ranging within limits prescribed in Table 1.

This is a good point, and could be interesting for testing the PE method. However, we would argue that knowledge of peat and mineral soil give us a priori knowledge that they will have very different properties. Knowing that we will always have at least two soil layers in tundra environments and, therefore, should include them in the inverse approach. So that land system models and other climate model would not use just one ground layer approach in they simulations.

**- L273-274: Let the reader know how you came to this conclusion.**

We added the following clarification:

"Figure 4 indicates that the method is able to recover porosities more robustly than thermal conductivities, i.e. estimated porosities are similar to their true state."

- Suggestion: I think that Figures 4 and 5 would be great in combination, e.g. the cost function as in Fig. 5 as an image (gray-scaled) in the background overlain by parameter trajectories on top as in Fig. 4. The same holds for Fig. 10 left and right.

Thank you for a great suggestion. We updated all Figures and implemented your suggestion for the corresponding figures.

- L296: Normalized to what? Maximum value? Please specify.

We added the following clarification:

"The color bar represents the cost function normalized by its maximum logarithmic value"

- L306: Why would one neglect the "outlier"? I thought the whole point of different starting models is an assessment of robustness? One out of five cannot be simply neglected.

That is true. However, in this case we are trying to compare all three cases shown on Figure 6 (now Figure 5). To clarify, we substitute it with the following text (L373-374).

Moreover, all except one estimated value showed a better match with its true value than the (S) $6T6s_l l\rho_a$  case without any added noise."

**- L417: "at least ten"? Why are there only five in Fig. 6 then? Please clarify.**

We removed that sentence from the current version of the manuscript. We did only five runs for the simplified meteorological data and then 10 for all other runs with actual met. data. For consistency we are showing only five points on each plot.

- L424-426: I do not feel that the term "accuracy" is appropriate here. The "data accuracy" does certainly not depend on the regularization technique. Please choose a different wording.

We changed that to robustness thought the manuscript.

- L426: "The regularization techniques". I suggest to leave out the "The" to be more general.

Changed.

- References: Please use a consistent formatting of citations (e.g., sometimes the year appears at the end, sometimes after the authors, sometimes in brackets, doi: vs. DOI: vs. https://doi.org/).

We made them all consistent. Now they all start with https://doi.org/

Technical corrections

- L76: "soil electrical resistances" -> "soil electrical resistivities". Done
- L100: "total number measurements" -> "the total number of measurements". Done
- L189: Remove "as if the synthetic truth is unknown". Done
- L365: Remove "by". Done
- L396: "matching" -> "estimating" (to differentiate between data and model space). Done

L427: "Its" -> "Their". Done
L454: Replace "confuse". Done
L462: "data" -> "parameters". Done
References

Tran, A. P., Dafflon, B., and Hubbard, S. S.: Coupled land surface–subsurface hydrogeophysical inverse modeling to estimate soil organic carbon content and explore associated hydrological and thermal dynamics in the Arctic tundra, The Cryosphere, 11, 2089-2109, https://doi.org/10.5194/tc-11-2089-2017, 2017.

**Reviewer #2**

I suggest the authors complete their description of model initialization by also specifying how the model domain is initialized in terms of initial temperature distribution (only the lower boundary fixed temperature is currently presented) and initial water saturation of the domain. It would also benefit the reader to know if the authors have conducted some sort of spin-up, to ensure that the model domain is in equilibrium with the forcing data.

We added the following lines clarifying the ATS model setup (L156-160):

"The setup of ATS models followed a standard procedure described in several studies (Atchley et al., 2015; Painter et al., 2016; Jafarov et al., 2018). Typically, we set up the model in several steps: 1) initialization of the water table, 2) introduction of the energy equation to establish antecedent permafrost, and 3) spinup of the model with simplified

and actual meteorological data from the BEO station. We spun up the model until the active layer cyclically equilibrated."

The authors define several abbreviations, some of which are used only few times. Please consider spelling out where abbreviation is not strictly necessary.

**We did, thank you.**

P5 Figure 1: Please expand the figure caption, and explain the parameters presented in the figure.

**"Figure 1.** Schematics of the parameter estimation algorithm. The algorithm starts with initial guesses on porosities and thermal conductivities  $\{\phi, k\}=\{\phi_m, \phi_p, k_m, k_p\}$  for the peat and mineral layers. The coupled ATS-BERT forward model then simulates temperature, liquid water, and apparent resistivities, which get passed to the cost function. If the cost function is small enough,  $\{\phi, k\}$  are considered to be the estimated parameters. If not, the values of the  $\{\phi, k\}$  are updated according to the Levenberg-Marquardt (LM) minimization algorithm and passed back to the ATS-BERT model."

P5 L123: if this is a transect through an ice-wedge polygon site, why are the ice wedges not represented in the model?

We added the following clarification at the beginning of section 2.1 (L137-141):

"To set up the synthetic model, we used digital elevation data of a transect through icewedge polygonal tundra at the Barrow Environmental Observatory (BEO), at Utqiagvik, Alaska (Fig. 2). Our study includes an 11 m section covering a single polygon with an ice-wedge on each side. In this study we do not explicitly assign ice properties for the ice-wedges. Instead, we model bulk porosities and effective thermal conductivities that can be associated with the entire transect."

L285-287: "The close proximity ...": very vague statement, please revise and elaborate. What is  $k_m$ ?

Now L393-397, in the current version we changed that sentence to:

"There is not much difference between cases (S) $3T3s_l l\rho_a$  and (S) $6T6s_l l\rho_a$  except for a slight improvement in  $k_m$ , suggesting that the small vertical distance (10 cm) between sensors 1 and 4, 2 and 5, and 3 and 6 could be limiting recorded data variability, leading to difficulty in estimation of the  $k_m$  parameter."

 $k_m$  is mineral layer thermal conductivity and we use this terminology more frequently now.

P11 L262: If each experiment was repeated 30 times, why does figure 6 show only 5 samples? and how were these samples selected from the full set? How many repetitions were used for the cases forced by observed meteorological data?

We added the following clarification at the beginning of section 3.1 (L325-330):

"We used 30 PE samples of  $\{\phi, k\}$  starting points in the first experiment ((S) $3T3s_l 1\rho_a$ ) to illustrate the overall performance of the parameter estimation using a large number of samples. After that, we did only five PE runs for the simplified meteorological data and 10 for all other runs with actual meteorological data. After Figure 4, for consistency and clarity, we show results for only five PE runs per case."

P12/13 figure 4&5: Which experiment does these figures represent? Why do the axes extend to physically meaningless (negative) values? Why are the axes limits in figure 5 cropped to exclude some of the minima found in the optimization (around k=1.4 W/(m.K) for mineral soil). The contour interval chosen for figure 5 is insufficient to visualize the optimization path leading to the outlier solutions.

We combined Figure 4 and 5. Now it is Figure 4. We added the experiment case to the caption and corrected the axis so that they do not extend to the non-physical area. We played with the number of contour lines in the plot. Increasing number of contour lines does not lead to improved contouring for the outliers, but makes the figure messier and harder to read. Here we choose and optimal number of contour lines that do not overload the figure.

P14 figure 6: Plots should be labeled with the full experiment designation, (S)6T6sl1r, (S)6T6sl1r+n and (S)6T6sl8r+n. Again, the plot axes extend to physically meaningless values.

We fixed all the issues with Figure 6, now 5.

Figures 7 and 8: The text in the tables and table titles is not readable P17 L365-366: "... increasing the number of measurement locations, leads to more accurate parameter estimation". I don't believe you have actually documented wether it is the number of measurement points, or the choice of location of the measurement points that cause the improvement. What would happen if you still used 6 measurement locations, but distributed differently (some deeper locations)?

We updated the Figures and removed all the numbers from the matrix tables and added the following text for more clarity (L398-391):

"We normalized the values in each matrix by dividing by the maximum value from the corresponding matrix. The normalized values are marked with tildes and range from 0 to 1, where values closer to 0 correspond to a better match and values closer to 1 correspond to a worse match."

We added an additional test Case #12, which showed a good performance and fell in Class I. We added corresponding additions on the importance of optimal sensor distribution within the transect in multiple places in the manuscript.

P18 L389-391: "In reality, the depth of the mineral soil..." this statement is unclear, please revise.

We changed that to the following (L446-448):

"At the actual site, the depth of the mineral soil can be deeper than 20 cm, not having sensors lower than 20 cm limits the amount of data that can help to improve the convexity of the cost function in our case."

P19 L397: misprint in matrix table reference, and wrong figure reference.

Fixed.

P21 L358-359: Again, you have not tested whether it is the location or number of observation points that matter.

Great suggestion. Now we added one more PE run (Case #12) that supports that statement.

P21 L475-476: "It is important to note that..." Unclear, please revise. Are you still referring to the ERT data?

We rewrote the entire conclusion in order to more closely relate it with results of the current study (see Comment #6).

P23 Figure 9: xlabel should read "Estimated properties accuracy" or similar.

We changed to "estimated parameter accuracy"

P24 Figure 10: Axes extend to physically meaningless values.

Fixed. Thank you.

---

## Referee Report (RR1)

[referee-annotated manuscript omitted]